# BiSSL: Bilevel Optimization for Self-Supervised Pre-Training and Fine-Tuning

## Abstract

In this work, we present BiSSL, a first-of-its-kind training framework that introduces bilevel optimization to enhance the alignment between the pretext pre-training and downstream fine-tuning stages in self-supervised learning. BiSSL formulates the pretext and downstream task objectives as the lower- and upper-level objectives in a bilevel optimization problem and serves as an intermediate training stage within the self-supervised learning pipeline. By more explicitly modeling the interdependence of these training stages, BiSSL facilitates enhanced information sharing between them, ultimately leading to a backbone parameter initialization that is better suited for the downstream task. We propose a training algorithm that alternates between optimizing the two objectives defined in BiSSL. Using a ResNet-18 backbone pre-trained with SimCLR on the STL10 dataset, we demonstrate that our proposed framework consistently achieves improved or competitive classification accuracies across various downstream image classification datasets compared to the conventional self-supervised learning pipeline. Qualitative analyses of the backbone features further suggest that BiSSL enhances the alignment of downstream features in the backbone prior to fine-tuning.

## 1 Introduction

In the absence of sufficient labeled data, self-supervised learning (SSL) has emerged as a promising approach for training deep learning models. Rather than relying solely on labeled data, the SSL framework aims to learn representations from unlabeled data which proves beneficial for subsequent use on various downstream tasks. These representations are learned by solving a pretext task, which utilizes supervisory signals extracted from the unlabeled data itself. Extensive efforts has gone into designing effective pretext tasks, achieving state-of-the-art or competitive performance in various fields such as computer vision (Chen et al., 2020b; Bardes et al., 2022; He et al., 2020; Grill et al., 2020; Caron et al., 2020; 2021; He et al., 2022; Oquab et al., 2024), audio signal processing (Schneider et al., 2019; Baevski et al., 2020; Hsu et al., 2021; Niizumi et al., 2021; Chung & Glass, 2018; Chung et al., 2019; Yadav et al., 2024) and natural language processing (Devlin et al., 2019; Lewis et al., 2019; Brown et al., 2020; He et al., 2021; Touvron et al., 2023).

Making a self-supervised pre-trained backbone suitable for a downstream task typically involves attaching additional layers that are compatible with that task, followed by fine-tuning the entire or parts of the composite model in a supervised manner (Zhai et al., 2019; Dubois et al., 2022). When a backbone is pre-trained on a distribution that differs from the distribution of the downstream data, the representations learned during pre-training may not be initially well-aligned with the downstream task. During fine-tuning, this distribution misalignment could cause relevant semantic information, learned during the pre-training phase, to vanish from the representation space (Zaiem et al., 2024; Chen et al., 2020a; Boschini et al., 2022). A potential strategy for alleviating the negative effects of these distribution discrepancies would be to enhance the alignment between the pretext pre-training and downstream fine-tuning stages. However, since the conventional SSL pipeline treats these stages as two disjoint processes, this poses a significant challenge in devising a strategy that enhances such alignment while not compromising on the benefits that SSL offers.

Meanwhile, bilevel optimization (BLO) has risen as a powerful tool for solving certain optimization problems within deep learning. It entails a main optimization problem constrained by the solution to a secondary optimization problem that depends on the parameters of the main objective. This hi-

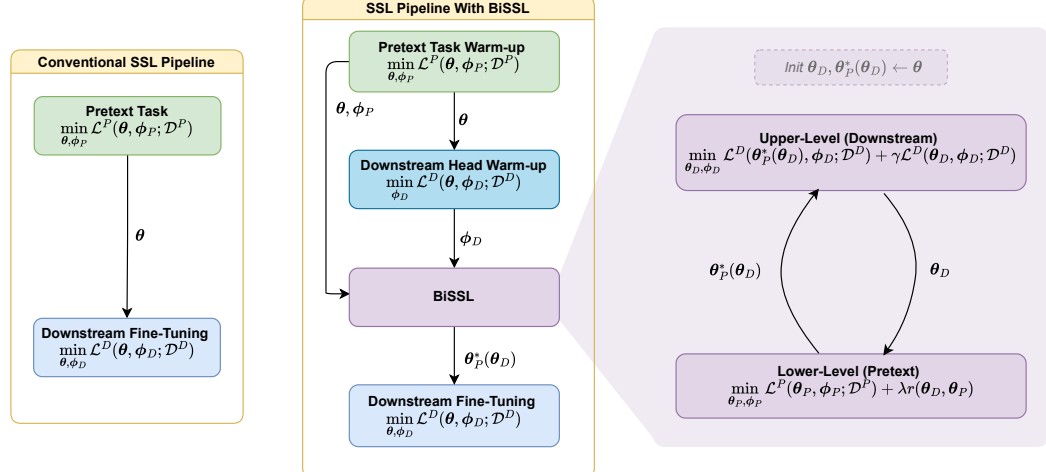

Figure 1: The conventional self-supervised learning pipeline alongside the proposed pipeline involving BiSSL. The symbols $\boldsymbol{\theta}$ and $\boldsymbol{\phi}$ represent obtained backbone and task-specific attached head parameters, respectively. When they are transmitted to the respective training stages, they are used as initializations.

erarchical setup causes the solutions of both optimization problems to depend on each other, either directly or implicitly, which has proven advantageous in deep learning tasks that optimize multiple inter-dependent objectives simultaneously (Zhang et al., 2023a). Notable mentions of tasks within deep learning where BLO has proven useful are parameter pruning (Zhang et al., 2022b), invariant risk minimization (Arjovsky et al., 2019; Zhang et al., 2023b), meta-learning (Rajeswaran et al., 2019; Finn et al., 2017), adversarial robustness (Zhang et al., 2021), hyper-parameter optimization (Franceschi et al., 2018) and coreset selection (Borsos et al., 2020).

In this study, we propose BiSSL, a novel training framework that leverages BLO to enhance the alignment between the pretext pre-training and downstream fine-tuning stages in SSL. Acting as an intermediate training stage within the SSL pipeline, BiSSL frames the pretext and downstream task objectives as the lower- and upper-level objectives in a BLO problem - a challenging approach that has not been explored until now. The objectives in BiSSL are connected by substituting the lower-level backbone solution for the upper-level backbone parameters, while simultaneously enforcing the lower-level backbone solution to resemble the upper-level backbone parameters. This approach more explicitly captures the interdependence between pretext pre-training and downstream fine-tuning, potentially leading to a lower-level backbone better aligned with the downstream task. Figure 1 compares the conventional SSL pipeline with our suggested pipeline involving BiSSL. Additionally, we propose a training algorithm for BiSSL and demonstrate that it consistently improves or maintains comparable downstream performance across a range of image classification datasets. For our experiments, we use SimCLR (Chen et al., 2020b) to pre-train a ResNet-18 backbone (He et al., 2016) on the unlabeled partition of the STL10 dataset (Coates et al., 2011), a setup offering suitable model capacity and dataset complexity while being less resource-intensive than larger-scale alternatives. The code implementation and pre-trained model weights are publicly available.[1]

## 2 RELATED WORK

**Bilevel Optimization in Self-Supervised Learning**    Bilevel optimization (BLO) refers to a constrained optimization problem, where the constraint itself is a solution to another optimization problem, which depends on the parameters of the "main" optimization problem. The general BLO

---

[1]https://github.com/ICLR25-10484/ICLR25_10484_BiSSL

problem is formulated as

$$\min_{\boldsymbol{\xi}} f(\boldsymbol{\xi}, \boldsymbol{\psi}^*(\boldsymbol{\xi})) \quad \text{s.t.} \quad \boldsymbol{\psi}^*(\boldsymbol{\xi}) \in \underset{\boldsymbol{\psi}}{\operatorname{argmin}}\, g(\boldsymbol{\xi}, \boldsymbol{\psi}), \tag{1}$$

where $f$ and $g$ are referred to as the upper-level and lower-level objectives, respectively. While the lower objective $g$ has knowledge of the parameters $\boldsymbol{\xi}$ from the upper-level objective, the upper-level objective $f$ possesses full information of the lower objective $g$ itself through its dependence on the lower-level solution $\boldsymbol{\psi}^*(\boldsymbol{\xi})$. Some works have incorporated bilevel optimization within self-supervised learning. Gupta et al. (2022) suggest formulating the contrastive self-supervised pretext task as a bilevel optimization problem, dedicating the upper-level and lower-level objectives for updating the backbone and projection head parameters respectively. Other frameworks such as the Local and Global (LoGo) (Zhang et al., 2022a) and Only Self-Supervised Learning (OSSL) Boonlia et al. (2022) utilize auxiliary models, wherein the lower-level objective optimizes the parameters of the auxiliary model, while the upper-level objective is dedicated to training the feature extraction model. MetaMask (Li et al., 2022b) introduces a meta-learning based approach, where the upper-level learns masks that filter out irrelevant information from inputs that are provided to a lower-level self-supervised contrastive pretext task. Chen et al. (2023) introduces a pseudo-BLO setup where the upper-level optimization still benefits from knowledge of the lower-level objective, but the parameters of the lower-level objective are fixed during training. In Somayajula et al. (2023), a two-staged BLO problem is proposed to fine-tune self-supervised pre-trained large language models in low-resource scenarios. Their approach focuses on solving downstream tasks while simultaneously learning a task-dependent similarity structure. BLO-SAM (Zhang et al., 2024) is tailored towards fine-tuning the segment anything model (SAM) (Kirillov et al., 2023) by interchangeably alternating between learning (upper-level) prompt embeddings and fine-tuning the (lower-level) segmentation model. The aforementioned frameworks integrate bilevel optimization into *either* the pre-training or fine-tuning stage exclusively and are tailored towards specific pretext or downstream tasks. In contrast, our proposed BiSSL employs a BLO problem that comprehensively incorporates *both* training stages of pretext pre-training and downstream fine-tuning, without being confined to any specific type of pretext or downstream task.

**Priming Pre-Trained Backbones Prior To Fine-Tuning** Previous works have demonstrated that downstream performance can be enhanced by introducing techniques that modify the backbone between the pre-training and fine-tuning stages. Contrastive Initialization (COIN) (Pan et al., 2022) introduces a supervised contrastive loss, to be utilised on backbones pre-trained with contrastive SSL techniques. Noisy-Tune (Wu et al., 2022) perturbs the pre-trained backbone with tailored noise before fine-tuning. Speaker-invariant clustering (Spin) (Chang et al., 2023) utilizes speaker disentanglement and vector quantization for improving speech representations for speech signal specific downstream tasks. RIFLE (Li et al., 2020) conducts multiple fine-tuning sessions sequentially, where the attached downstream specific layers are re-initialized in between every session. Unlike BiSSL, these techniques either do not incorporate knowledge of both the pretext task and downstream task objectives and their relationship or do so only implicitly.

## 3 PROPOSED METHOD

### 3.1 NOTATION

We denote the unlabeled pretext dataset $\mathcal{D}^P = \{\mathbf{z}_k\}_{k=1}^{C_P}$ and labeled downstream dataset $\mathcal{D}^D = \{\mathbf{x}_l, \mathbf{y}_l\}_{k=1}^{C_D}$, respectively, where $\mathbf{z}_k, \mathbf{x}_l \in \mathbb{R}^N$. Let $f_{\boldsymbol{\theta}} : \mathbb{R}^N \to \mathbb{R}^M$ denotes a feature extracting backbone with trainable parameters $\boldsymbol{\theta}$ and $p_{\boldsymbol{\phi}} : \mathbb{R}^M \to \mathbb{R}^P$ a task specific projection head with trainable parameters $\boldsymbol{\phi}$. Given pretext and downstream models $g_{\boldsymbol{\phi}_P} \circ f_{\boldsymbol{\theta}_P}$ and $h_{\boldsymbol{\phi}_D} \circ f_{\boldsymbol{\theta}_D}$ with $\boldsymbol{\theta}_P, \boldsymbol{\theta}_D \in \mathbb{R}^L$, we denote the pretext and downstream training objectives $\mathcal{L}^P(\boldsymbol{\theta}_P, \boldsymbol{\phi}_P; \mathcal{D}^P)$ and $\mathcal{L}^D(\boldsymbol{\theta}_D, \boldsymbol{\phi}_D; \mathcal{D}^D)$, respectively. To simplify notation, we omit the dataset specification from the training objectives, e.g. $\mathcal{L}^D(\boldsymbol{\theta}_D, \boldsymbol{\phi}_D) := \mathcal{L}^D(\boldsymbol{\theta}_D, \boldsymbol{\phi}_D; \mathcal{D}^D)$.

### 3.2 OPTIMIZATION PROBLEM FORMULATION

The conventional setup of self-supervised pre-training directly followed by supervised fine-tuning relies on using a single backbone model with parameters $\boldsymbol{\theta}$. In that instance, we minimize

$\mathcal{L}^P(\boldsymbol{\theta}, \boldsymbol{\phi}_P)$ to produce a backbone parameter configuration $\boldsymbol{\theta}^*$ which is then used as an initialization when subsequently minimizing the downstream training objective $\mathcal{L}^D(\boldsymbol{\theta}, \boldsymbol{\phi}_D)$. We deviate from this by instead considering $\boldsymbol{\theta}_P$ and $\boldsymbol{\theta}_D$ as two separate parameter vectors that are strongly correlated. In continuation, we suggest combining the two traditionally separate optimization problems of pretext and downstream training into a joint optimization problem through bilevel optimization called BiSSL. We formulate BiSSL as

$$\min_{\boldsymbol{\theta}_D, \boldsymbol{\phi}_D} \quad \mathcal{L}^D\left(\boldsymbol{\theta}_P^*\left(\boldsymbol{\theta}_D\right), \boldsymbol{\phi}_D\right) + \gamma \mathcal{L}^D\left(\boldsymbol{\theta}_D, \boldsymbol{\phi}_D\right) \tag{2}$$

$$\text{s.t.} \quad \boldsymbol{\theta}_P^*\left(\boldsymbol{\theta}_D\right) \in \underset{\boldsymbol{\theta}_P}{\operatorname{argmin}} \min_{\boldsymbol{\phi}_P} \mathcal{L}^P\left(\boldsymbol{\theta}_P, \boldsymbol{\phi}_P\right) + \lambda r(\boldsymbol{\theta}_D, \boldsymbol{\theta}_P) \tag{3}$$

with $\gamma \in \mathbb{R}_+$ and $r$ being some convex regularisation objective weighted by $\lambda \in \mathbb{R}_+$ enforcing similarity between $\boldsymbol{\theta}_D$ and $\boldsymbol{\theta}_P$. The upper-level training objective in equation 2 is tasked with minimizing the downstream task objective $\mathcal{L}^D$, while the lower-level objective in equation 3 aims to minimize the pretext task objective $\mathcal{L}^P$ while also ensuring its backbone remains similar to the upper-level backbone. As seen in the left term of equation 2, the backbone parameters $\boldsymbol{\theta}_P^*(\boldsymbol{\theta}_D)$ are transferred into the downstream training objective, mirroring how the backbone is transferred in the conventional SSL pipeline. Although the second term of equation 2 is not strictly necessary, it has empirically shown to improve stability and aid convergence of the upper-level solution during training. Unlike the traditional SSL setup, the backbone solution of the pretext objective $\boldsymbol{\theta}_P^*(\boldsymbol{\theta}_D)$ is now a function of the parameters of the downstream backbone $\boldsymbol{\theta}_D$, as the lower-level problem is dependent on the upper-level backbone parameters.

As the upper-level objective in equation 2 depends on the solution $\boldsymbol{\theta}_P^*(\boldsymbol{\theta}_D)$ of the lower-level objective in equation 3, this enables the incorporation of information from the pretext objective when solving the upper-level optimization problem. By including a regularization objective $r$ that enforces similarity between the lower-level and upper-level backbone parameters, this setup is hypothesized to guide the lower-level to achieve a configuration of model backbone parameters that is more beneficial for subsequent conventional fine-tuning on the downstream task. To more precisely understand how the pretext objective influences the downstream training procedure in this setup, we delve deeper into the expression of the gradient of the upper-level training objective in equation 2 in the following subsection.

### 3.3 UPPER-LEVEL DERIVATIVE

Given the upper-level objective $F(\boldsymbol{\theta}_D, \boldsymbol{\phi}_D) := \mathcal{L}^D(\boldsymbol{\theta}_P^*(\boldsymbol{\theta}_D), \boldsymbol{\phi}_D) + \gamma \mathcal{L}^D(\boldsymbol{\theta}_D, \boldsymbol{\phi}_D)$ from equation 2, its derivative with respect to $\boldsymbol{\theta}_D$ is given by

$$\frac{\mathrm{d}F}{\mathrm{d}\boldsymbol{\theta}_D} = \underbrace{\frac{\mathrm{d}\boldsymbol{\theta}_P^*(\boldsymbol{\theta}_D)}{\mathrm{d}\boldsymbol{\theta}_D}^T}_{\text{IG}} \nabla_{\boldsymbol{\theta}}\mathcal{L}^D(\boldsymbol{\theta}, \boldsymbol{\phi}_D)|_{\boldsymbol{\theta}=\boldsymbol{\theta}_P^*(\boldsymbol{\theta}_D)} + \gamma \nabla_{\boldsymbol{\theta}}\mathcal{L}^D(\boldsymbol{\theta}, \boldsymbol{\phi}_D)|_{\boldsymbol{\theta}=\boldsymbol{\theta}_D}. \tag{4}$$

Due to the dependence of the lower-level solution on the upper-level parameters, the first term of equation 4 includes the implicit gradient (IG) of the implicit function $\boldsymbol{\theta}_P^*(\boldsymbol{\theta}_D)$. To simplify notation, we let $\nabla_{\boldsymbol{\xi}}h(\boldsymbol{\xi})|_{\boldsymbol{\xi}=\boldsymbol{\psi}} := \nabla_{\boldsymbol{\xi}}h(\boldsymbol{\psi})$ when it is clear from context which variables are differentiated with respect to. Following an approach similar to Rajeswaran et al. (2019), with details on the derivations and underlying assumptions outlined in Section A.1 of Appendix A, the IG in equation 4 can be explicitly expressed as

$$\frac{\mathrm{d}\boldsymbol{\theta}_P^*(\boldsymbol{\theta}_D)}{\mathrm{d}\boldsymbol{\theta}_D}^T = -\nabla^2_{\boldsymbol{\theta}_D \boldsymbol{\theta}_P} r(\boldsymbol{\theta}_D, \boldsymbol{\theta}_P^*(\boldsymbol{\theta}_D)) \left[\nabla^2_{\boldsymbol{\theta}}\left(\frac{1}{\lambda}\mathcal{L}^P(\boldsymbol{\theta}_P^*(\boldsymbol{\theta}_D), \boldsymbol{\phi}_P) + r(\boldsymbol{\theta}_D, \boldsymbol{\theta}_P^*(\boldsymbol{\theta}_D))\right)\right]^{-1}. \tag{5}$$

A common convex regularization objective, which will also be the choice in the subsequent experiments of this work, is $r(\boldsymbol{\xi}, \boldsymbol{\psi}) = \frac{1}{2}\|\boldsymbol{\xi} - \boldsymbol{\psi}\|_2^2$. Using this regularization objective simplifies equation 5 down to

$$\frac{\mathrm{d}\boldsymbol{\theta}_P^*(\boldsymbol{\theta}_D)}{\mathrm{d}\boldsymbol{\theta}_D}^T = \left[\frac{1}{\lambda}\nabla^2_{\boldsymbol{\theta}}\mathcal{L}^P(\boldsymbol{\theta}_P^*(\boldsymbol{\theta}_D), \boldsymbol{\phi}_P) + I_L\right]^{-1}, \tag{6}$$

where $I_L$ is the $L \times L$-dimensional identity matrix. Hence the upper-level derivative in equation 4 can be expressed as

$$\frac{\mathrm{d}F}{\mathrm{d}\boldsymbol{\theta}_D} = \left[\frac{1}{\lambda}\nabla^2_{\boldsymbol{\theta}}\mathcal{L}^P(\boldsymbol{\theta}_P^*(\boldsymbol{\theta}_D), \boldsymbol{\phi}_P) + I_L\right]^{-1} \nabla_{\boldsymbol{\theta}}\mathcal{L}^D(\boldsymbol{\theta}_P^*(\boldsymbol{\theta}_D), \boldsymbol{\phi}_D) + \gamma \nabla_{\boldsymbol{\theta}}\mathcal{L}^D(\boldsymbol{\theta}_D, \boldsymbol{\phi}_D). \tag{7}$$

The inverse Hessian-vector product in the left term of equation 7 is computationally infeasible to calculate directly, hence it is approximated using the conjugate gradient (CG) method (Nazareth, 2009; Shewchuk, 1994). While CG is established as a successful approach for approximating the inverse Hessian-vector products in previous works (Pedregosa, 2016; Zhang et al., 2021; Rajeswaran et al., 2019), it still introduces significant computational overhead due to its need for iterative evaluations of multiple Hessian vector products. Future work may explore alternative methods that offer more efficient approximations without compromising downstream task performance. We employ a layer-wise implementation of the CG method based on that of Rajeswaran et al. (2019) and refer to their work for more details on applying CG in a deep learning setup with BLO. For a comprehensive overview of other common methods used to approximate the upper-level derivative in BLO, we refer to Zhang et al. (2023a).

With an explicit expression of the IG in equation 6, we can interpret the impact of the scaling factor $\lambda$ from equation 3 and equation 7: When $\lambda$ is very large, the dependence of lower-level objective on the upper-level parameters $\boldsymbol{\theta}_D$ is also very large. This effectively drives the lower-level backbone parameters toward the trivial solution $\boldsymbol{\theta}_P^*(\boldsymbol{\theta}_D) = \boldsymbol{\theta}_D$. Meanwhile, the IG in equation 6 approximately equals $I_L$, thereby diminishing the influence of the lower-level objective on the upper-level gradient in equation 7. This roughly makes the task of the upper-level equivalent to conventional fine-tuning. Conversely, if $\lambda$ is very small, the lower-level objective in equation 3 effectively defaults to conventional pretext task training. Additionally, the implicit gradient in equation 6 would consist of numerically tiny entries, making the optimization of the first term in the upper-level objective in equation 2 equivalent to probing of the downstream head on the frozen pretext backbone $\boldsymbol{\theta}_P^*(\boldsymbol{\theta}_D)$.

## 3.4 TRAINING ALGORITHM AND PIPELINE

---

**Algorithm 1** BiSSL Training Algorithm

---

1: **Input:** Backbone and projection head parameter initializations $\boldsymbol{\theta}$, $\boldsymbol{\phi}_P$, $\boldsymbol{\phi}_D$. Training objectives $\mathcal{L}^P$, $\mathcal{L}^D$. Weights $\lambda, \gamma \in \mathbb{R}_+$. Optimizers $\text{opt}_P$, $\text{opt}_D$. Number of training stage alternations $T \in \mathbb{N}$ with upper and lower-level iterations $N_U, N_L \in \mathbb{N}$. Upper-level backbone adaption frequency $N_a \in \mathbb{N}$ and strength $\alpha \in [0, 1]$.

2: Initialize $\boldsymbol{\theta}_P \leftarrow \boldsymbol{\theta}$ and $\boldsymbol{\theta}_D \leftarrow \boldsymbol{\theta}$.

3: **for** $t = 1, \ldots, T$ **do**
4:     **for** $n = 1, \ldots, N_L$ **do**                           ▷ Lower-level
5:         Compute $\mathbf{g}_{\boldsymbol{\phi}_P} = \nabla_{\boldsymbol{\phi}} \mathcal{L}^P(\boldsymbol{\theta}_P, \boldsymbol{\phi})|_{\boldsymbol{\phi}=\boldsymbol{\phi}_P}$ .
6:         Compute $\mathbf{g}_{\boldsymbol{\theta}_P} = \nabla_{\boldsymbol{\theta}} \mathcal{L}^P(\boldsymbol{\theta}, \boldsymbol{\phi}_P)|_{\boldsymbol{\theta}=\boldsymbol{\theta}_P} + \lambda \nabla_{\boldsymbol{\theta}} r(\boldsymbol{\theta}_D, \boldsymbol{\theta})|_{\boldsymbol{\theta}=\boldsymbol{\theta}_P}$.
7:         Update $\boldsymbol{\phi}_P \leftarrow \text{opt}_P(\boldsymbol{\phi}_P, \mathbf{g}_{\boldsymbol{\phi}_P})$ and $\boldsymbol{\theta}_P \leftarrow \text{opt}_P(\boldsymbol{\theta}_P, \mathbf{g}_{\boldsymbol{\theta}_P})$.

8:     **if** $t \bmod N_a \equiv 0$ **then**
9:         $\boldsymbol{\theta}_D \leftarrow (1 - \alpha)\boldsymbol{\theta}_D + \alpha \boldsymbol{\theta}_P$.

10:     **for** $n = 1, \ldots, N_U$ **do**                         ▷ Upper-level
11:         Compute $\mathbf{g}_{\boldsymbol{\phi}_D} = \nabla_{\boldsymbol{\phi}} \mathcal{L}^D(\boldsymbol{\theta}_P, \boldsymbol{\phi})|_{\boldsymbol{\phi}=\boldsymbol{\phi}_D} + \gamma \nabla_{\boldsymbol{\phi}} \mathcal{L}^D(\boldsymbol{\theta}_D, \boldsymbol{\phi})|_{\boldsymbol{\phi}=\boldsymbol{\phi}_D}$.
12:         Compute $\mathbf{v} = \nabla_{\boldsymbol{\theta}} \mathcal{L}^D(\boldsymbol{\theta}, \boldsymbol{\phi}_D)|_{\boldsymbol{\theta}=\boldsymbol{\theta}_P}$.
13:         Approximate $\mathbf{v}_{\text{IG}} \approx \left[ I_M + \frac{1}{\lambda} \nabla_{\boldsymbol{\theta}}^2 \mathcal{L}^P(\boldsymbol{\theta}, \boldsymbol{\phi}_P)|_{\boldsymbol{\theta}=\boldsymbol{\theta}_P} \right]^{-1} \mathbf{v}$.         ▷ Use CG
14:         Compute $\mathbf{g}_{\boldsymbol{\theta}_D} = \mathbf{v}_{\text{IG}} + \gamma \nabla_{\boldsymbol{\theta}} \mathcal{L}^D(\boldsymbol{\theta}, \boldsymbol{\phi}_D)|_{\boldsymbol{\theta}=\boldsymbol{\theta}_D}$.
15:         Update $\boldsymbol{\phi}_D \leftarrow \text{opt}_D(\boldsymbol{\phi}_D, \mathbf{g}_{\boldsymbol{\phi}_D})$ and $\boldsymbol{\theta}_D \leftarrow \text{opt}_D(\boldsymbol{\theta}_D, \mathbf{g}_{\boldsymbol{\theta}_D})$.

16: **Return:** Backbone Parameters $\boldsymbol{\theta}_P$.

---

Algorithm 1 outlines the proposed training algorithm, which iteratively alternates between solving the lower-level (equation 3) and upper-level (equation 2) optimization problems in BiSSL. The lower-level training optimizes the pretext task objective, while additionally including the gradient of the regularization term $r$ for the backbone parameter updates, complying with equation 3. For the upper-level training, the gradient with respect to the backbone parameters as represented by the left term on the right-hand side in equation 7, is approximated using the CG method. Additionally, the

pretext backbone parameters $\boldsymbol{\theta}_P$ are weighted by $\alpha$ and added to the downstream backbone parameters $\boldsymbol{\theta}_D$ every $N_a$ alternations to further enforce similarity between them, which empirically has shown to aid convergence during training.

From Section A.1 in Appendix A, we get that $\boldsymbol{\theta}_P^*(\boldsymbol{\theta}_D)$ must fulfill the stationary condition $\nabla_{\boldsymbol{\theta}}\big(\mathcal{L}^P(\boldsymbol{\theta}, \boldsymbol{\phi}_P) + \lambda r(\boldsymbol{\theta}_D, \boldsymbol{\theta})\big)|_{\boldsymbol{\theta}=\boldsymbol{\theta}_P^*(\boldsymbol{\theta}_D)} = \mathbf{0}$ to justify the explicit expression of the implicit gradient in equation 6. This means that executing Algorithm 1 using random initializations of $\boldsymbol{\theta}$ and $\boldsymbol{\phi}_P$ will likely not suffice. The same applies to $\boldsymbol{\phi}_D$, as a random initialization of $\boldsymbol{\phi}_D$ typically leads to rapid initial changes of the backbone parameters $\boldsymbol{\theta}_D$ during fine-tuning. This would then likely violate the assumed stationary condition due to the dependence between $\boldsymbol{\theta}_D$ and $\boldsymbol{\theta}_P$ through the regularization objective $r$. Figure 1 illustrates the suggested pipeline alongside the conventional SSL pipeline. First conventional pretext pre-training is performed on the unlabeled dataset $\mathcal{D}^P$ to obtain initializations of $\boldsymbol{\theta}$ and $\boldsymbol{\phi}_P$. Next, the downstream head is fitted on top of the frozen backbone $\boldsymbol{\theta}$ using the downstream dataset $\mathcal{D}^D$, which provides an initialization of the downstream head parameters $\boldsymbol{\phi}_D$. Then, BiSSL training is conducted as outlined in Algorithm 1, yielding an updated configuration of backbone parameters $\boldsymbol{\theta}_P^*(\boldsymbol{\theta}_D)$. These updated backbone parameters are subsequently used as an initialization for the final supervised fine-tuning on the downstream task.

## 4 EXPERIMENTS AND RESULTS

### 4.1 DATASETS

The STL10 dataset (Coates et al., 2011) is used throughout the experiments. It comprises two partitions: 100.000 unlabeled images and 13.000 labeled images with 10 classes in total whereas 5000 and 8000 are assigned for training and testing, respectively. All images are natural images of resolution $96 \times 96$, with the unlabeled partition drawn from a similar but broader distribution than the labeled partition. This dataset strikes a balance between complexity and computational feasibility, offering higher resolution and more diverse content than smaller datasets like CIFAR10 (Krizhevsky, 2012) while being less resource-intensive than larger-scale datasets such as ImageNet (Deng et al., 2009). For ease of reference, STL10U and STL10L will denote the unlabeled and labeled partitions, respectively. In all experiments, STL10U will be employed for self-supervised pre-training. For downstream fine-tuning and evaluation, we leverage a varied set of natural image classification datasets that encompass a wide array of tasks, including general image classification, fine-grained recognition across species and objects, scene understanding, and texture categorization. The datasets include STL10L, Oxford 102 Flowers (Nilsback & Zisserman, 2008), StanfordCars (Yang et al., 2015), FGVC Aircraft (Maji et al., 2013), Describable Textures Dataset (DTD) (Cimpoi et al., 2014), Oxford-IIIT Pets (Parkhi et al., 2012), FashionMNIST (Xiao et al., 2017), CIFAR10 (Krizhevsky, 2012), CIFAR100 (Krizhevsky, 2012), Caltech-101 (Li et al., 2022a), Food 101 (Bossard et al., 2014), SUN397 scene dataset (Xiao et al., 2010), Caltech-UCSD Birds-200-2011 (CUB200) (Wah et al., 2011) and PASCAL VOC 2007 (Everingham et al.). All downstream datasets are split into training, validation, and test partitions, with details on how these assignments are made provided in Section B.1 of Appendix B.

### 4.2 IMPLEMENTATION DETAILS

#### 4.2.1 BASELINE SETUP

**Pretext Task Training** The SimCLR (Chen et al., 2020b) pretext task with temperature $\tau = 0.5$ is used for pre-training a ResNet-18 (He et al., 2016) backbone model. We selected this widely adopted architecture due to its proven ability to extract high-quality visual representations while maintaining relatively low computational requirements, striking an effective balance between performance and resource efficiency. On top of the backbone, a projection head is used, consisting of two fully connected layers with batch normalization (Ioffe & Szegedy, 2015) and ReLU (Agarap, 2018) followed by a single linear layer. Each layer consists of 256 neurons.

The image augmentation scheme follows the approach used in Bardes et al. (2022), with minor modifications: The image size is set to $96 \times 96$ instead of $224 \times 224$, and the minimal ratio of the random crop is adjusted accordingly to 0.5 instead of 0.08.

The implementation of the LARS optimizer (You et al., 2017) from Bardes et al. (2022) is employed, with a "trust" coefficient of $0.001$, a weight decay of $10^{-6}$ and a momentum of $0.9$. The learning rate increases linearly during the first 10 epochs, reaching a peak base learning rate of $4.8$, followed by a cosine decay towards 0 with no restarts (Loshchilov & Hutter, 2017) for the remaining epochs. A batch size of $1024$ is used and, unless otherwise specified, pre-training is conducted for 600 epochs.

**Fine-Tuning on the Downstream Task**   For downstream fine-tuning, a single linear layer is attached to the output of the pre-trained backbone. The training procedure utilizes the cross-entropy loss, the SGD optimizer with a momentum of $0.9$, and a cosine decaying learning rate scheduler without restarts (Loshchilov & Hutter, 2017). Fine-tuning is conducted for 400 epochs with a batch size of 256. An augmentation scheme similar to the fine-tuning augmentation scheme in Bardes et al. (2022) is employed, where images are center cropped and resized to $96 \times 96$ pixels with a minimal crop ratio of $0.5$, followed by random horizontal flips.

A random grid search of 200 hyper-parameter configurations for the learning rates and weight decays is conducted, where one model is fine-tuned for each configuration. Base learning rates and weight decays are log-uniformly sampled over the ranges of $0.0001$ to $1.0$ and $0.00001$ to $0.01$, respectively. Validation data accuracy is evaluated after each epoch. The hyper-parameter configuration yielding the best balance between high validation accuracy and low validation loss is considered the optimal hyper-parameter configuration.[2] The corresponding optimal hyper-parameters for each downstream dataset are documented in Table 2 of Appendix B.

For subsequent evaluation on the test data, we train 10 models with different random seeds, each using the considered optimal hyper-parameter configurations. During the training of each respective model, the model parameters are stored after each epoch if the top-1 validation accuracy (or 11-point mAP for the VOC07 dataset) has increased compared to the previous highest top-1 validation accuracy achieved during training. Top-1 and top-5 test data accuracies (or 11-point mAP for the VOC07 dataset) are evaluated for each of the 10 models, from which the calculated means and standard deviations of these accuracies are documented.

### 4.2.2   BiSSL Setup

In this section, we detail each stage of the proposed training pipeline for BiSSL, as outlined in the right part of Figure 1.

**Pretext Warm-up**   The backbone $\theta$ and projection head $\phi_P$ are initialized by self-supervised pre-training using a setup almost identical to the baseline pretext task training setup in Section 4.2.1. The only difference is that this training stage is conducted for 500 epochs instead of 600 epochs, and that the peak base learning rate is set to $1.0$ instead of $4.8$. This adjustment is made because the BiSSL training stage will conduct what is roughly equivalent to 100 pretext epochs, as detailed more specifically in the composite configuration paragraph below. This ensures that the total number of pretext pre-training steps is comparable to those conducted in the baseline setup.

**Downstream Head Warm-up**   The training setup for the downstream head warm-up closely mirrors the fine-tuning setup of Section 4.2.1. The main difference is that only the linear downstream head is fitted on top of the now frozen backbone obtained from the pretext warm-up. Learning rates and weight decays are initially selected based on those listed in Table 2, with adjustments made as needed when preliminary testing indicated a potential for improved convergence. These values are provided in Table 3 in Appendix B. The authors recognize that more optimal hyper-parameter configurations may exist and leave further exploration of this for future refinement. The downstream head warm-up is conducted for 20 epochs with a constant learning rate.

**Lower-level of BiSSL**   The training configuration for the lower-level primarily follows the setup described for pretext pre-training in Section 4.2.1, with the modifications outlined here. As specified in equation 3, the lower-level loss function is the sum of the pretext task objective $\mathcal{L}^P$ (in our case, the NT-Xent loss from SimCLR (Chen et al., 2020b)) and the regularization term $r(\boldsymbol{\theta}_D, \boldsymbol{\theta}_P) =$

---

[2]In certain scenarios during the experiments, the configuration that achieved the highest validation accuracy also yielded a notably higher relative validation loss. To ensure better generalizability, an alternative configuration with a more favorable trade-off was selected in these cases.

Table 1: Test classification accuracies. Accuracies significantly different from their counterparts are marked in bold font.

| Dataset | Top-1 Accuracy (*: 11-point mAP) | | | Top-5 Accuracy | | |
|---|---|---|---|---|---|---|
| | BiSSL | Only FT | Avg Diff | BiSSL | Only FT | Avg Diff |
| STL10L | $90.2 \pm 0.1$ | $90.3 \pm 0.1$ | $-0.1$ | $\mathbf{99.7 \pm 0.0}$ | $99.6 \pm 0.0$ | $\mathbf{+0.1}$ |
| Flowers | $\mathbf{74.8 \pm 0.2}$ | $73.4 \pm 0.4$ | $\mathbf{+1.4}$ | $89.8 \pm 0.3$ | $90.0 \pm 0.4$ | $-0.2$ |
| Cars | $73.0 \pm 0.4$ | $72.7 \pm 0.5$ | $+0.3$ | $91.5 \pm 0.3$ | $91.4 \pm 0.4$ | $+0.1$ |
| Aircrafts | $46.9 \pm 0.5$ | $46.1 \pm 0.9$ | $+0.8$ | $78.9 \pm 0.4$ | $79.3 \pm 0.6$ | $-0.4$ |
| DTD | $\mathbf{51.8 \pm 0.5}$ | $49.3 \pm 0.5$ | $\mathbf{+2.5}$ | $\mathbf{79.9 \pm 0.3}$ | $79.1 \pm 0.4$ | $\mathbf{+0.8}$ |
| Pets | $\mathbf{67.8 \pm 0.2}$ | $65.0 \pm 0.5$ | $\mathbf{+2.8}$ | $\mathbf{92.3 \pm 0.3}$ | $90.7 \pm 0.3$ | $\mathbf{+1.6}$ |
| FMNIST | $94.3 \pm 0.2$ | $94.1 \pm 0.1$ | $+0.2$ | $100.0 \pm 0.0$ | $100.0 \pm 0.0$ | $0.0$ |
| CIFAR10 | $93.9 \pm 0.1$ | $93.8 \pm 0.1$ | $+0.1$ | $\mathbf{99.9 \pm 0.0}$ | $99.8 \pm 0.0$ | $\mathbf{+0.1}$ |
| CIFAR100 | $73.0 \pm 0.1$ | $73.2 \pm 0.2$ | $-0.2$ | $\mathbf{93.7 \pm 0.1}$ | $92.8 \pm 0.1$ | $\mathbf{+0.9}$ |
| Caltech-101 | $\mathbf{80.6 \pm 0.7}$ | $78.1 \pm 0.5$ | $\mathbf{+2.5}$ | $\mathbf{95.5 \pm 0.2}$ | $94.7 \pm 0.2$ | $\mathbf{+0.8}$ |
| Food | $72.0 \pm 0.2$ | $71.7 \pm 0.2$ | $+0.3$ | $90.4 \pm 0.1$ | $90.4 \pm 0.1$ | $0.0$ |
| SUN397 | $\mathbf{41.1 \pm 0.2}$ | $40.0 \pm 0.3$ | $\mathbf{+1.1}$ | $\mathbf{71.0 \pm 0.2}$ | $69.9 \pm 0.4$ | $\mathbf{+1.1}$ |
| CUB200 | $\mathbf{47.1 \pm 0.4}$ | $45.7 \pm 0.4$ | $\mathbf{+1.4}$ | $\mathbf{72.1 \pm 0.3}$ | $70.7 \pm 0.6$ | $\mathbf{+1.4}$ |
| VOC07 | $*\mathbf{60.4 \pm 0.1}$ | $*58.6 \pm 0.3$ | $\mathbf{+1.8}$ | $-$ | $-$ | $-$ |

$\frac{1}{2}||\boldsymbol{\theta}_D - \boldsymbol{\theta}_P||_2^2$. Based on early experiments, the regularization weight $\lambda = 0.001$ was selected, as it appeared to strike a well-balanced compromise between the convergence rates of both the lower- and upper-level objectives. The lower-level is trained for the equivalent of approximately 100 conventional pre-training epochs, with further details provided in the composite configuration paragraph. Each time the BiSSL training alternates back to the lower-level, the first 5 batches used for lower-level training are stored. These stored batches are utilized to approximate the Hessian of the lower-level objective when approximating the upper-level gradient. Further details are specified in Section B.3 of Appendix B and the paragraph below.

**Upper-level of BiSSL**  The upper-level training stage also shares many similarities with the downstream training setup described in Section 4.2.1, and again, only the differences are addressed here. The weight decays and base learning rates are set to match those obtained from the downstream head warm-up detailed in Table 3 of Appendix B. The weighting of the conventional downstream loss objective is set to $\gamma = 0.01$. To approximate the upper-level gradient in equation 7, the conjugate gradient method (Nazareth, 2009; Shewchuk, 1994) is employed. Further details regarding the setup for the upper-level gradient approximation are covered in Section B.3 of Appendix B.

**Composite Configuration Details of BiSSL**  As outlined in Algorithm 1, both lower- and upper-level backbone parameters $\boldsymbol{\theta}_P$ and $\boldsymbol{\theta}_D$ are initialized with the backbone parameters obtained during the pretext warm-up, and the training procedure alternates between solving the lower- and upper-level optimization problems. In this experimental setup, the lower-level performs $N_L = 20$ gradient steps before alternating to the upper-level, which then conducts $N_U = 8$ gradient steps. A total of $T = 500$ training stage alternations are executed. As the STL10U dataset with the current batch size of 1024 amounts to a total of 98 training batches without replacement, these $T = 500$ training stage alternations roughly equal 100 conventional pretext epochs. Section B.4 in Appendix B outlines further details on how data batches are handled during training. The upper-level backbone adaptation frequency and strength are set to $N_a = 100$ and $\alpha = 0.1$, respectively. Additionally, gradient normalization is employed on gradients exceeding $\ell_2$-norms of 10.

**Fine-Tuning on the Downstream Task**  Subsequent downstream fine-tuning is conducted in a manner identical to that described in the 'Fine-Tuning on the Downstream Task" paragraph of section 4.2.1. Table 4 in Appendix B lists the considered optimal hyper-parameter configurations for each dataset.

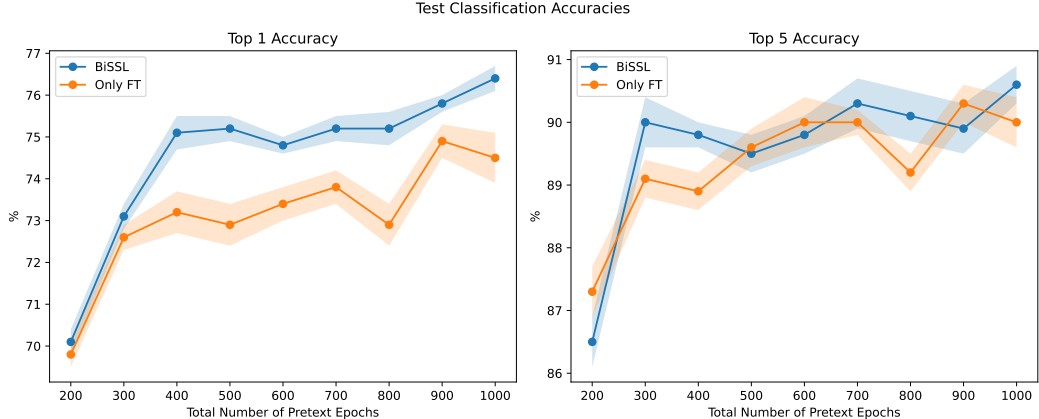

Figure 2: Test classification accuracies on the Flowers dataset for separate models pre-trained for different durations, comparing the conventional and BiSSL training pipelines. BiSSL consistently achieves higher top-1 accuracy than the baseline after sufficient pre-training.

### 4.3 DOWNSTREAM TASK PERFORMANCE

The impact of using BiSSL compared to the conventional self-supervised training pipeline is benchmarked by evaluating classification accuracies on the various specified downstream datasets. Table 1 presents the means and standard deviations of top-1 and top-5 classification accuracies (or the 11-point mAP on the VOC2007 dataset) on these downstream test datasets, comparing results obtained from the conventional SSL pipeline with those achieved using the BiSSL pipeline. The results demonstrate that training with BiSSL significantly improves either top-1 or top-5 classification accuracy, or 11-point mAP in the case of VOC07, on 10 out of 14 datasets, with no single result showing a significant decline in performance compared to the baseline.

#### 4.3.1 PERFORMANCE OVER VARYING PRE-TRAINING EPOCHS

To further assess the robustness of BiSSL, we conduct experiments varying the duration of self-supervised pre-training while keeping the remaining experimental setup unchanged. The Flowers dataset, which demonstrated substantial benefits from BiSSL in terms of top-1 classification accuracies, was chosen for these experiments. We continue to compare models with BiSSL applied after 100 fewer pre-training epochs than the baseline, ensuring the total pretext training duration is approximately equal across methods, as described in Section 4.2.2. Figure 2 depicts the final fine-tuned test accuracy achieved by separate models pre-trained for varying durations. The results indicate that BiSSL consistently outperforms the baseline once a sufficient duration of pre-training is reached. This suggests that the BiSSL's benefits are not contingent on the amount of pre-training, but rather that it provides a more efficient learning trajectory, stemming from the enhanced information sharing it facilitates between the pretext and downstream tasks.

### 4.4 VISUAL INSPECTION OF LATENT FEATURES

To gain deeper insight into how BiSSL affects the representations learned compared to conventional pretext pre-training, we perform a qualitative visual inspection of latent spaces. This involves comparing features processed by backbones trained solely by pretext pre-training to those derived from lower-level backbones obtained after conducting BiSSL, each trained as described in the "Pretext Task Training" and "Lower-level of BiSSL" paragraphs in Section 4.2.1 and 4.2.2, respectively. By comparing these features, we aim to assess whether BiSSL nudges the latent features toward being more semantically meaningful for the downstream task. The t-Distributed Stochastic Neighbor Embedding (t-SNE) (Cieslak et al., 2020) technique is employed for dimensionality reduction. Further details regarding the experimental setup are outlined in Section C.1 of Appendix C. Figure 3 illustrates the results on the flowers dataset, indicating that BiSSL yields backbones with improved

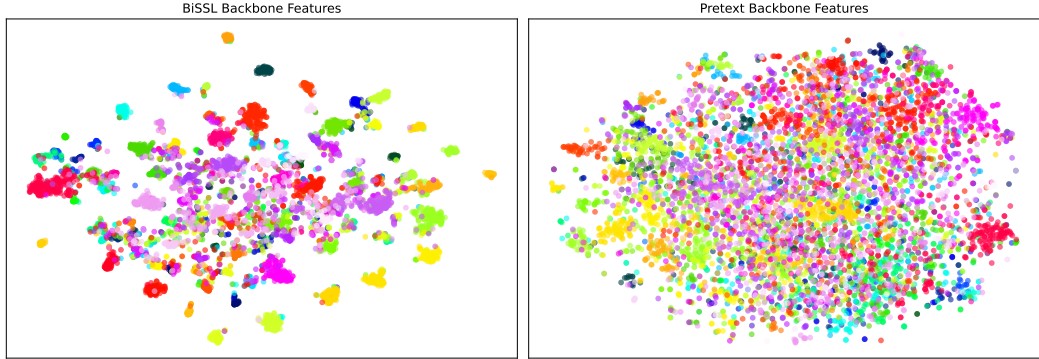

Figure 3: Visualization of features from backbones trained using pretext pre-training exclusively and backbones derived from lower-level backbones obtained after applying BiSSL, respectively. Features are extracted from the test partition of the flowers dataset. Each color represents a different class. Details are outlined in Section C.1 of Appendix C

downstream feature alignment. Further plots on a selection of downstream datasets in Section C.1 reinforce this finding, also demonstrating that this trend persists even for datasets where BiSSL did not impose any classification accuracy improvements.

# 5 CONCLUSION

This study integrates pretext pre-training and downstream fine-tuning into a unified bilevel optimization problem, from which the BiSSL training framework is proposed. BiSSL explicitly models the inheritance of backbone parameters from the pretext task, enhancing the transfer of relevant information between the pretext and downstream tasks. We propose a practical training algorithm and pipeline that incorporates BiSSL as an intermediate stage between pretext pre-training and downstream fine-tuning. Experiments across various image classification datasets demonstrate that BiSSL consistently achieves improved or comparable downstream classification performance relative to the conventional self-supervised learning pipeline. Additionally, our findings indicate that in instances where BiSSL improves performance, this improvement remains consistent regardless of the pretext pre-training duration. Further analysis suggests that BiSSL enhances the downstream semantic richness of learned representations, as evidenced by qualitative inspections of latent spaces. BiSSL marks a potential advancement towards enhancing the alignment between the pretext pre-training and downstream fine-tuning stages, revealing a new direction for self-supervised learning algorithm designs that leverage bilevel optimization.

## 5.1 FUTURE WORK

Formulating the self-supervised pipeline as a bilevel optimization problem offers various strategies with trade-offs in computational complexity and theoretical justification. While this study presents a promising approach for improving downstream performance, further investigation of alternative formulations is needed to identify setups are are potentially more optimal. Although BiSSL is theoretically applicable to any downstream task and model size, our experiments focused on small-scale image classification due to resource constraints. Therefore, it remains uncertain whether BiSSL can scale to larger setups and tasks. Additionally, a potential future advancement would integrating more novel methods for solving BLO problems, which promise benefits in terms of reduced computational costs and improved solution convergence (Zhang et al., 2023a; Yang et al., 2021; Choe et al., 2023; Huang, 2024). Lastly, the current BiSSL framework relies on full access to pre-training data and pretext tasks. Future research could investigate the use of only a subset of pre-training data and alternative pretext tasks to maintain BiSSL's benefits under these conditions.

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

## A  THEORETICAL INSIGHTS AND FRAMEWORK COMPARISONS IN BiSSL

### A.1  DERIVATION OF THE IMPLICIT GRADIENT

Assume the setup of the BiSSL optimization problem described in equation 2 and equation 3. In the following derivations, we will assume that $\phi_P$ is fixed, allowing us to simplify the expressions involved. To streamline the notation further, we continue to use the convention $\nabla_{\boldsymbol{\xi}} h(\boldsymbol{\xi})|_{\boldsymbol{\xi}=\boldsymbol{\psi}} := \nabla_{\boldsymbol{\xi}} h(\boldsymbol{\psi})$, when it is clear from context which variables are differentiated with respect to. Under these circumstances, we then define the lower-level objective from equation 3 as

$$G(\boldsymbol{\theta}_D, \boldsymbol{\theta}_P) := \mathcal{L}^P(\boldsymbol{\theta}_P, \boldsymbol{\phi}_P) + \lambda r(\boldsymbol{\theta}_D, \boldsymbol{\theta}_P). \tag{8}$$

Recalling that $r$ is a *convex* regularization objective, adequate scaling of $\lambda$ effectively "convexifies" the lower-level objective $G$, a strategy also employed on the lower-level objective in previous works (Rajeswaran et al., 2019; Zhang et al., 2022b; 2023a). This is advantageous because assuming convexity of $G$ ensures that for any $\boldsymbol{\theta}_D \in \mathbb{R}^L$, there exists a corresponding $\hat{\boldsymbol{\theta}}_P \in \mathbb{R}^L$ that satisfies the stationary condition $\nabla_{\boldsymbol{\theta}_P} G(\boldsymbol{\theta}_D, \hat{\boldsymbol{\theta}}_P) = \mathbf{0}$. In other words, we are assured that a minimizer of $G(\boldsymbol{\theta}_D, \cdot)$ exists for all $\boldsymbol{\theta}_D \in \mathbb{R}^L$. Now, further assume that $\nabla_{\boldsymbol{\theta}_P} G(\boldsymbol{\theta}_D, \boldsymbol{\theta}_P)$ is continuously differentiable and that the Hessian matrix $\nabla^2_{\boldsymbol{\theta}_P} G(\boldsymbol{\theta}_D, \hat{\boldsymbol{\theta}}_P)$ is invertible for all $\boldsymbol{\theta}_D \in \mathbb{R}^L$. Under these conditions, the implicit function theorem (Dontchev & Rockafellar, 2014; Zucchet & Sacramento, 2022) guarantees the existence of an implicit unique and *differentiable* function $\boldsymbol{\theta}_P^* : \mathcal{N}(\boldsymbol{\theta}_D) \to \mathbb{R}^L$, with $\mathcal{N}(\boldsymbol{\theta}_D)$ being a neighborhood of $\boldsymbol{\theta}_D$, that satisfies $\boldsymbol{\theta}_P^*(\boldsymbol{\theta}_D) = \hat{\boldsymbol{\theta}}_P$ and $\nabla_{\boldsymbol{\theta}_P} G(\tilde{\boldsymbol{\theta}}_D, \boldsymbol{\theta}_P^*(\tilde{\boldsymbol{\theta}}_D)) = \mathbf{0}$ for all $\tilde{\boldsymbol{\theta}}_D \in \mathcal{N}(\boldsymbol{\theta}_D)$.

As the lower-level solution $\boldsymbol{\theta}_P^*(\boldsymbol{\theta}_D)$ is indeed a differentiable function under these conditions, this justifies that the expression

$$\frac{\mathrm{d}}{\mathrm{d}\boldsymbol{\theta}_D} \nabla_{\boldsymbol{\theta}_P} [G(\boldsymbol{\theta}_D, \boldsymbol{\theta}_P^*(\boldsymbol{\theta}_D))] = \mathbf{0}$$

is valid for all $\boldsymbol{\theta}_D \in \mathbb{R}^L$. By applying the chain rule, the expression becomes

$$\nabla^2_{\boldsymbol{\theta}_D \boldsymbol{\theta}_P} G(\boldsymbol{\theta}_D, \boldsymbol{\theta}_P^*(\boldsymbol{\theta}_D)) + \frac{\mathrm{d}\boldsymbol{\theta}_P^*(\boldsymbol{\theta}_D)}{\mathrm{d}\boldsymbol{\theta}_D}^T \nabla^2_{\boldsymbol{\theta}_P} G(\boldsymbol{\theta}_D, \boldsymbol{\theta}_P^*(\boldsymbol{\theta}_D)) = \mathbf{0}.$$

Recalling that $\nabla^2_{\boldsymbol{\theta}_P} G(\boldsymbol{\theta}_D, \boldsymbol{\theta}_P^*(\boldsymbol{\theta}_D))$ is assumed to be invertible, the IG $\frac{\mathrm{d}\boldsymbol{\theta}_P^*(\boldsymbol{\theta}_D)}{\mathrm{d}\boldsymbol{\theta}_D}^T$ can be isolated

$$\frac{\mathrm{d}\boldsymbol{\theta}_P^*(\boldsymbol{\theta}_D)}{\mathrm{d}\boldsymbol{\theta}_D}^T = -\nabla^2_{\boldsymbol{\theta}_D \boldsymbol{\theta}_P} G(\boldsymbol{\theta}_D, \boldsymbol{\theta}_P^*(\boldsymbol{\theta}_D)) \left[ \nabla^2_{\boldsymbol{\theta}_P} G(\boldsymbol{\theta}_D, \boldsymbol{\theta}_P^*(\boldsymbol{\theta}_D)) \right]^{-1},$$

and by substituting the expression for $G$ from equation 8, the expression becomes

$$\frac{\mathrm{d}\boldsymbol{\theta}_P^*(\boldsymbol{\theta}_D)}{\mathrm{d}\boldsymbol{\theta}_D}^T = -\nabla^2_{\boldsymbol{\theta}_D \boldsymbol{\theta}_P} r(\boldsymbol{\theta}_D, \boldsymbol{\theta}_P^*(\boldsymbol{\theta}_D)) \left[ \nabla^2_{\boldsymbol{\theta}_P} \left( \frac{1}{\lambda} \mathcal{L}^P(\boldsymbol{\theta}_P^*(\boldsymbol{\theta}_D), \boldsymbol{\phi}_P) + r(\boldsymbol{\theta}_D, \boldsymbol{\theta}_P^*(\boldsymbol{\theta}_D)) \right) \right]^{-1}. \tag{9}$$

**To summarize**, given the following assumptions:

- The lower-level pretext projection head parameters $\boldsymbol{\phi}_P$ are fixed.
- $G$ is convex such that $\nabla_{\boldsymbol{\theta}_P} G(\boldsymbol{\theta}_D, \boldsymbol{\theta}_P^*(\boldsymbol{\theta}_D)) = \mathbf{0}$ is fulfilled for every $\boldsymbol{\theta}_D \in \mathbb{R}^L$.
- The Hessian matrix $\nabla^2_{\boldsymbol{\theta}_P} G(\boldsymbol{\theta}_D, \boldsymbol{\theta}_P^*(\boldsymbol{\theta}_D))$ exists and is invertible for all $\boldsymbol{\theta}_D \in \mathbb{R}^L$.

Then, the IG $\frac{\mathrm{d}\boldsymbol{\theta}_P^*(\boldsymbol{\theta}_D)}{\mathrm{d}\boldsymbol{\theta}_D}^T$ can be explicitly expressed by equation 9. The authors acknowledge that an explicit expression for the IG without fixing $\boldsymbol{\phi}_P$ is achievable, though this is left for future exploration.

### A.2  DISTINCTION FROM BILEVEL OPTIMIZATION IN META-LEARNING

While bilevel optimization (BLO) has been applied in meta-learning frameworks such as MAML (Finn et al., 2017), Sign-MAML (Fan et al., 2021) and iMAML (Rajeswaran et al., 2019),

BiSSL represents a distinct application and implementation of BLO, tailored for the challenges of self-supervised learning (SSL). In the aforementioned works, BLO is primarily utilized to address few-shot learning scenarios, focusing on efficiently adapting models to new tasks with minimal labeled data. Conversely, BiSSL applies BLO to concurrently manage the more complex task of self-supervised pretext pre-training on unlabeled with downstream fine-tuning on labeled data. Another key distinction is that in meta-learning, the upper- and lower-level objectives are closely related, with the upper-level objective formulated as a summation of the lower-level tasks. In contrast, BiSSL involves fundamentally distinct objectives at each level, utilizing separate datasets and tasks for pre-training and fine-tuning. This design allows BiSSL to better align the pre-trained model with the requirements of a specific downstream task. Conversely, the BLO in meta-learning aims to broadly generalize across a wide range of tasks, prioritizing adaptability rather than task-specific optimization. Additionally, unlike BiSSL, the meta-learning frameworks discussed reinitialize the lower-level backbone parameters with a copy of the upper-level parameters at every iteration. In BiSSL, the closest comparable mechanism is the occasional update of the upper-level backbone using an EMA update with the lower-level parameters (see Algorithm 1), though this occurs far less frequently.

## B  EXPERIMENTAL DETAILS

### B.1  DATASET PARTITIONS

The Caltech-101 (Li et al., 2022a) dataset does not come with a pre-defined train/test split, so the same convention as previous works is followed (Chen et al., 2020b; Donahue et al., 2014; Simonyan & Zisserman, 2014), where 30 random images per class are selected for the training partition, and the remaining images are assigned for the test partition. For the DTD (Cimpoi et al., 2014) and SUN397 (Xiao et al., 2010) datasets, which offer multiple proposed train/test partitions, the first splits are used, consistent with the approach in Chen et al. (2020b).

For downstream hyperparameter optimization, portions of the training partitions from each respective labeled dataset are designated as validation datasets. The FGVC Aircraft (Maji et al., 2013), Oxford 102 Flowers (Nilsback & Zisserman, 2008), DTD, and Pascal VOC 2007 (Everingham et al.) datasets already have designated validation partitions. For all the remaining labeled datasets, the validation data partitions are randomly sampled while ensuring that class proportions are maintained. For the multi-attribute VOC07 dataset, sampling is performed with class balance concerning the first attribute present in each image. Roughly $20\%$ of the training data is allocated for validation.

### B.2  DOWNSTREAM TASK FINE-TUNING OF THE BASELINE SETUP

In Table 2, the learning rates and weight decays used for each respective downstream dataset of the experiments described in Section 4.2.1 are outlined.

### B.3  DOWNSTREAM HEAD WARMUP AND UPPER-LEVEL OF BiSSL

Table 3 outlines the learning rates and weight decays used for the downstream head warm-up and upper-level of BiSSL of each respective downstream dataset, as described in the BiSSL experimental setup of Section 4.2.2.

The first term of the upper-level gradient equation 7 is approximated using the Conjugate Gradient (CG) method (Nazareth, 2009; Shewchuk, 1994). Our implementation follows a similar structure to that used in Rajeswaran et al. (2019), employing $N_c = 5$ iterations and a dampening term $\lambda_{\mathrm{damp}} = 10$. Given matrix $A$ and vector $\mathbf{v}$, the CG method iteratively approximates $A^{-1}\mathbf{v}$, which requires evaluation of multiple matrix-vector products $A\mathbf{d}_1, \ldots, A\mathbf{d}_{N_c}$. In practice, storing the matrix $A$ (in our case, the Hessian $\nabla^2_{\boldsymbol{\theta}_P} \mathcal{L}^P(\boldsymbol{\theta}_P^*(\boldsymbol{\theta}_D), \boldsymbol{\phi}_P)$) in its full form is typically infeasible. Instead, a function that efficiently computes the required matrix-vector products instead of explicitly storing the matrix is typically utilized. For our setup, this function is detailed in Algorithm 2, showing how the $K$ stored lower-level batches (we use $K = 5$ as previously outlined in the "Lower-level of BiSSL" paragraph in Section 4.2.2) are used to calculate Hessian-vector products. This approach ensures that the output of the CG algorithm is an approximation of the inverse Hessian-vector product in the first term of Equation equation 7 as intended.

Table 2: Hyper-parameter configurations used for downstream fine-tuning after conventional pretext pre-training yielding the highest top-1 classification accuracies (11-point mAP for the VOC2007 dataset).

| Dataset | Learning Rate | Weight Decay |
|---|---|---|
| STL10L | 0.0136 | 0.001 |
| Flowers | 0.113 | 0.00226 |
| Cars | 0.035 | 0.00658 |
| Aircrafts | 0.0167 | 0.00996 |
| DTD | 0.0262 | 0.00332 |
| Pets | 0.0235 | 0.00472 |
| FashionMNIST | 0.0009 | 0.00829 |
| CIFAR10 | 0.0067 | 0.00128 |
| CIFAR100 | 0.005 | 0.00127 |
| Caltech-101 | 0.0096 | 0.00902 |
| Food | 0.015 | 0.00699 |
| SUN397 | 0.0097 | 0.00121 |
| CUB200 | 0.0722 | 0.00568 |
| VOC2007 | 0.0108 | 0.00894 |

Table 3: Hyper-parameters used for the Downstream Head Warm-up and Upper-level of BiSSL.

| Dataset | Learning Rate | Weight Decay |
|---|---|---|
| STL10L | 0.015 | 0.01 |
| Flowers | 0.05 | 0.01 |
| Cars | 0.035 | 0.007 |
| Aircrafts | 0.015 | 0.01 |
| DTD | 0.015 | 0.0075 |
| Pets | 0.03 | 0.005 |
| FashionMNIST | 0.05 | 0.004 |
| CIFAR10 | 0.03 | 0.006 |
| CIFAR100 | 0.03 | 0.001 |
| Caltech-101 | 0.03 | 0.007 |
| Food | 0.015 | 0.01 |
| SUN397 | 0.03 | 0.002 |
| CUB200 | 0.05 | 0.005 |
| VOC2007 | 0.03 | 0.006 |

## B.4 COMPOSITE CONFIGURATION OF BiSSL

To avoid data being reshuffled between every training stage alternation, the respective batched lower- and upper-level training datasets are stored in separate stacks from which data is drawn. The stacks are only "reset" when the number of remaining batches is smaller than the number of gradient steps required before alternating to the other level. For example, the lower-level stack is reshuffled every fourth training stage alternation. If the downstream dataset does not provide enough data for making $N_U = 8$ batches with non-overlapping data points, the data is simply reshuffled every time

---

**Algorithm 2** Hessian Vector Product Calculation $f_H$ (To use in the CG Algorithm)

---

1: **Input:** Input vector $\mathbf{v}$. Model parameters $\boldsymbol{\theta}_P$, $\boldsymbol{\phi}_P$. Training objective $\mathcal{L}^P$. Lower-level data batches $[\mathbf{z}_1, \ldots, \mathbf{z}_K]$. Regularization weight $\lambda$ and dampening $\lambda_{\mathrm{damp}}$.

2: Initialize $\mathbf{y} \leftarrow \mathbf{0}$    ▷ Initialize Hessian vector product $\mathbf{y}$
3: **for** $k = 1, \ldots, K$ **do**
4:    $\pi(\boldsymbol{\theta}_P) \leftarrow \left(\nabla_{\boldsymbol{\theta}} \mathcal{L}^P\left(\boldsymbol{\theta}, \boldsymbol{\phi}_P; \mathbf{z}_k\right)\big|_{\boldsymbol{\theta}=\boldsymbol{\theta}_P}\right)^T \mathbf{v}$
5:    $\mathbf{g} \leftarrow \nabla_{\boldsymbol{\theta}} \pi(\boldsymbol{\theta})\big|_{\boldsymbol{\theta}=\boldsymbol{\theta}_P}$    ▷ Memory efficient calculation of $\nabla_{\boldsymbol{\theta}}^2 \mathcal{L}^P(\boldsymbol{\theta}, \boldsymbol{\phi}_P; \mathbf{z}_k)|_{\boldsymbol{\theta}=\boldsymbol{\theta}_P}\mathbf{x}$.
6:    $\mathbf{y} \leftarrow \mathbf{y} + \frac{1}{K}\mathbf{g}$
7: $\mathbf{y} \leftarrow \mathbf{v} + \frac{1}{\lambda + \lambda_{\mathrm{damp}}}\mathbf{y}$

8: **Return:** $f_H(\mathbf{v}) := \mathbf{y}$

---

the remaining number of data points is smaller than the upper-level batch size (256 images in these experiments).

### B.5 DOWNSTREAM FINE-TUNING AFTER BiSSL

The learning rates and weight decays used for downstream fine-tuning after BiSSL for each respective downstream dataset are outlined in Table 4. Section 4.2.2 outlines the experimental setup.

Table 4: Hyper-parameter configurations used for downstream fine-tuning after BiSSL leading to the highest top-1 classification accuracies (11-point mAP for the VOC2007 dataset).

| Dataset | Learning Rate | Weight Decay |
|---|---|---|
| STL10L | 0.0005 | 0.00043 |
| Flowers | 0.0009 | 0.00005 |
| Cars | 0.0293 | 0.00851 |
| Aircrafts | 0.011 | 0.00612 |
| DTD | 0.0008 | 0.00764 |
| Pets | 0.0002 | 0.00543 |
| FashionMNIST | 0.0022 | 0.00876 |
| CIFAR10 | 0.0003 | 0.00991 |
| CIFAR100 | 0.0012 | 0.00422 |
| Caltech-101 | 0.0008 | 0.00011 |
| Food | 0.006 | 0.0095 |
| SUN397 | 0.0011 | 0.00028 |
| CUB200 | 0.035 | 0.00868 |
| VOC2007 | 0.0002 | 0.00015 |

## C ADDITIONAL RESULTS

### C.1 VISUAL INSPECTION OF LATENT FEATURES

Test data features of the downstream test data processed by backbones trained through conventional pretext pre-training are compared against those trained with BiSSL. This allows for an inspection of the learned representations prior to the final fine-tuning stage.

During the evaluation, it is important to note that the batch normalization layers (Ioffe & Szegedy, 2015) of the pre-trained backbones utilize the running mean and variance inferred during train-

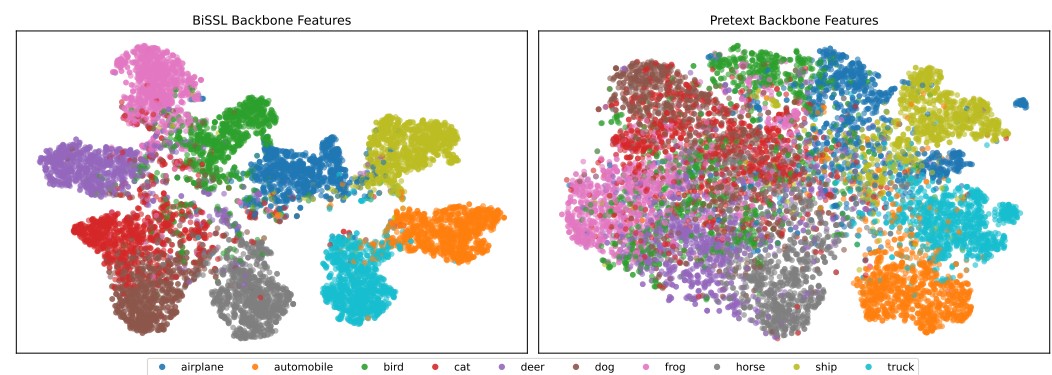

Figure 4: CIFAR10

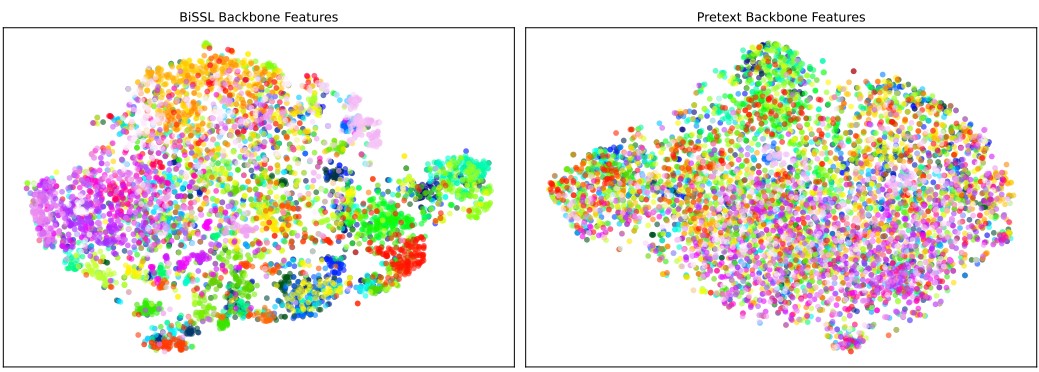

Figure 5: CUB200

ing. Since these pre-trained backbones have not been exposed to the downstream datasets during training, their batch normalization statistics may not be optimal for these new datasets. To address this, the training dataset is divided into batches of 256 samples, and roughly 100 batches are then forward-passed through the backbones. This procedure ensures that the batch normalization statistics are better suited to the downstream datasets, thereby providing a fairer comparison of the learned representations.

For the dimensionality reduction and visualization of these latent features, the t-Distributed Stochastic Neighbor Embedding (t-SNE) (Cieslak et al., 2020) technique is employed. This method allows us to visually assess the clustering and separation of features in the latent space, providing qualitative insights into the semantic structure of the representations learned through BiSSL.

Figures 3 to 11 illustrate the outcomes of these visual inspections on a selection of the downstream datasets described in Section 4.1, highlighting the differences in feature representations between conventional pretext pre-training and BiSSL.

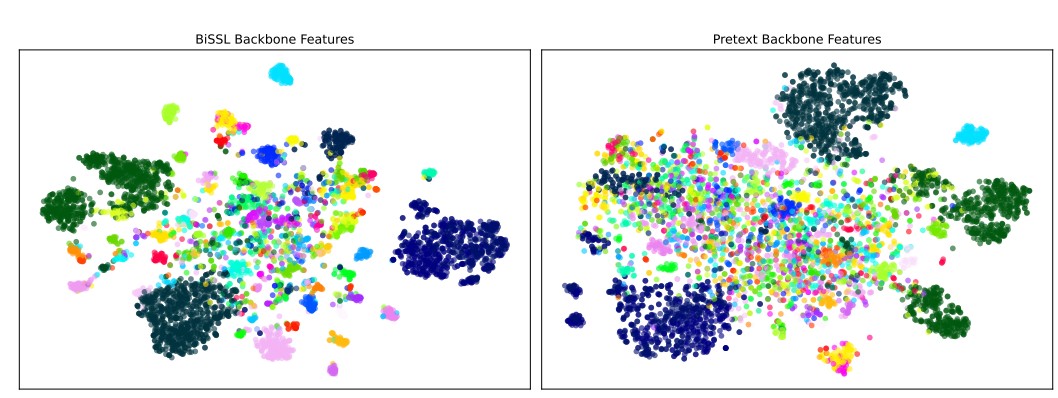

Figure 6: Caltech-101

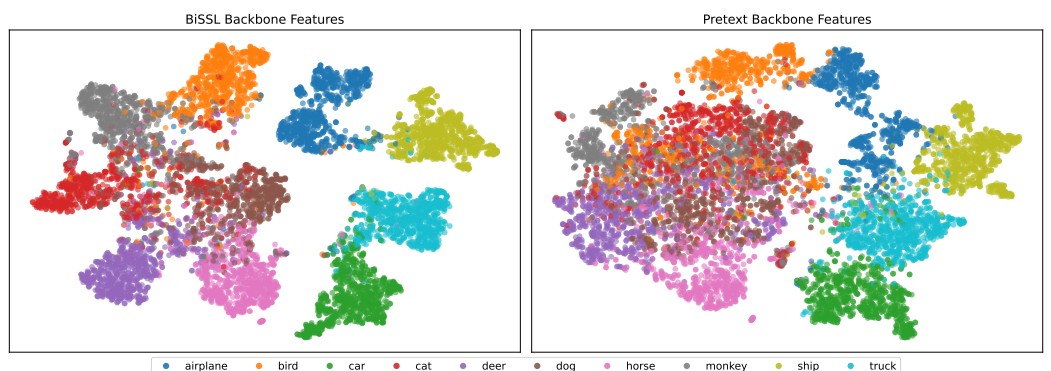

Figure 7: STL10L

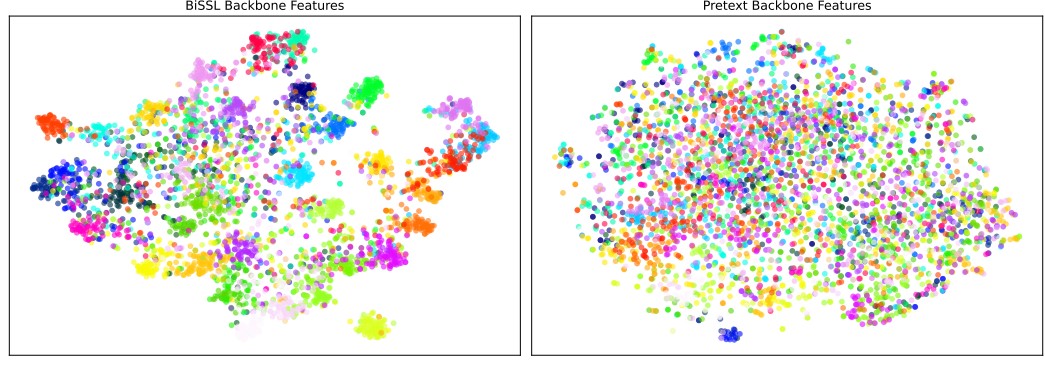

Figure 8: Pets

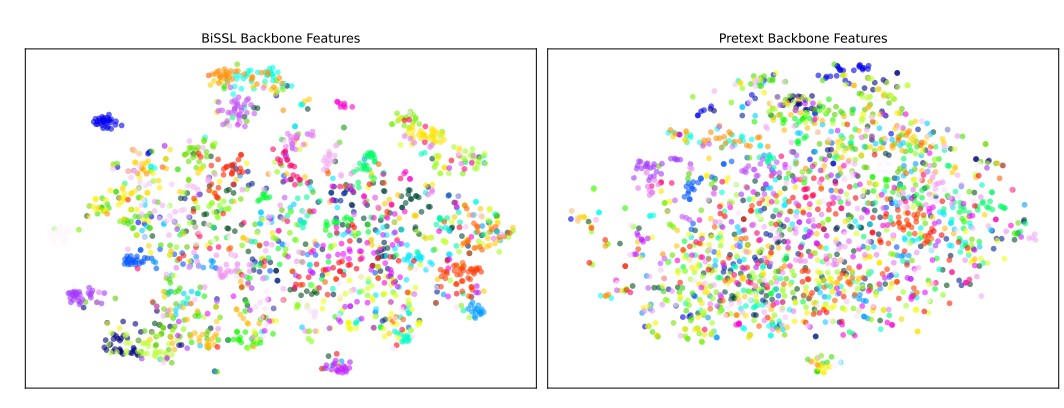

Figure 9: DTD

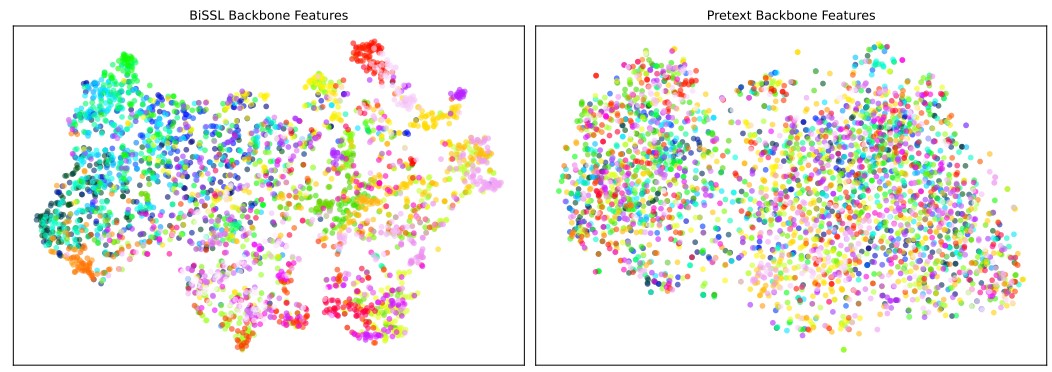

Figure 10: Aircrafts

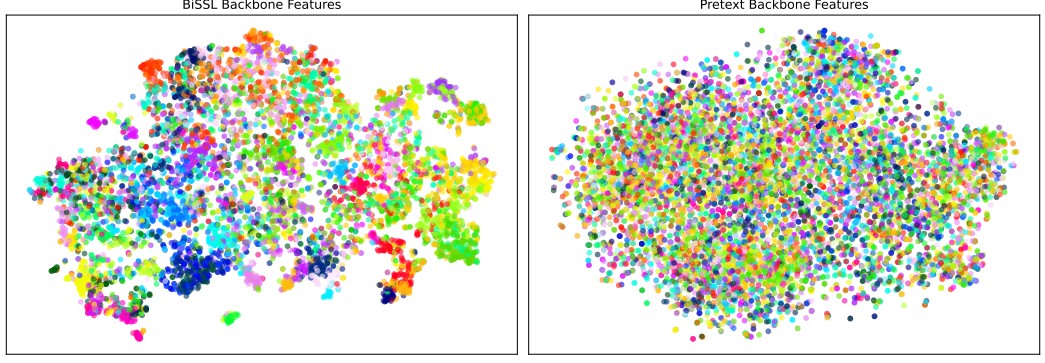

Figure 11: Cars

