# OpenReview forum: "BiSSL: Bilevel Optimization for Self-Supervised Pre-Training and Fine-Tuning"
_ICLR.cc/2025/Conference — Submitted to ICLR 2025_

### Official Review · Reviewer_ZY6B · 2024-10-27

**Soundness:** 3
**Presentation:** 3
**Contribution:** 3
**Rating:** 6
**Confidence:** 3

**Summary:**

This paper proposes a framework called BiSSL, which is a bilevel optimization framework for self-supervised pre-training and fine-tuning. It enhances the alignment between pre-training and fine-tuning. It serves as an intermediate training stage in the self-supervised learning pipeline. It also enhances information sharing between the pretext and fine-tuning stage. As a result, the backbone initialization can be more suitable for downstream tasks. Experiment results prove the effectiveness of the proposed method.

**Strengths:**

1. This paper is overall well-written and easy to read, with a clear organization. The figures and algorithm illustrates the idea very clearly.
2. The idea of introducing bilevel optimization into pre-training and fine-tuning pipeline is reasonable for me.
3. The experiment design is good, covering both quantitative and qualitative results, together with some analysis.

**Weaknesses:**

1. As mentioned in strengths, the idea of this paper is reasonable, but not very new to me. The bilevel optimization is widely used in meta-learning.
2. In the experiment section, the backbone (ResNet-18) seem to be relatively smaill.

**Questions:**

1. I think if possible, the authors can have more discussions on the novelty of this paper.
2. I am curious about the results of the proposed method in larger backbone (at least ResNet-50).
3. The authors can have more discussions on the limitation of the proposed method.

---

> ### Author Response · Authors · 2024-11-20
> **Author's Response to Reviewer ZY6B [Part 1/2]**
>
> We thank the reviewer for their constructive comments. We address them as follows:
>
> ---
>
> ## Strengths
> > 1. *This paper is overall well-written and easy to read, with a clear organization. The figures and algorithm illustrates the idea very clearly.*
>
> We are pleased that the reviewer found the paper well-structured and appreciated the clarity of the figures and algorithm.
>
> > 2. *The idea of introducing bilevel optimization into pre-training and fine-tuning pipeline is reasonable for me.*
> > 3. *The experiment design is good, covering both quantitative and qualitative results, together with some analysis.*
>
> It is delighting to see that the reviewer finds the integration of bilevel optimization into this context to be reasonable and we are grateful for the recognition of our evaluation!
>
> ---
>
> ## Weaknesses
> > 1. *As mentioned in strengths, the idea of this paper is reasonable, but not very new to me. The bilevel optimization is widely used in meta-learning.*
>
> While BLO has been utilized in meta-learning frameworks such as MAML [1] and iMAML [2], we would like to emphasize that our method and application differ from these approaches in several key aspects. In meta-learning, BLO is used to address few-shot learning tasks, where the focus is on adapting models to new tasks with minimal data. As present in our work, however, we leverage BLO to tackle the more complex task of SSL pre-training on a vast unlabeled dataset along with fine-tuning concurrently. Another important distinction is that in meta-learning, the upper and lower-level objectives are often closely related, with the upper level typically being a summation of the lower-level tasks. In contrast, our framework involves entirely different objectives at each level, each also using potentially fundamentally different datasets. Moreover, BiSSL is tailored to fine-tune pre-trained backbone models, in contrast to the conventional use of BLO in meta-learning, which primarily focuses on training model parameters from an uninitialized state.
>
> We have added a section in the appendix of the paper to include a clarification on how our approach differs from meta-learning, based on the response provided above.
>
> > 2. *In the experiment section, the backbone (ResNet-18) seem to be relatively smaill.*
>
> While we acknowledge that ResNet-18 is a relatively small-scale model by current standards, its use demonstrates an applicable aspect for scenarios that that do not rely on relatively large architectures. However, we agree that exploring larger backbones is important to align with modern trends in SSL. To this end, we conducted additional experiments using ResNet-50 backbone. We refer to our response to question 2 for results and details.
>
> *Continued...*

---

> ### Author Response · Authors · 2024-11-20
> **Author's Response to Reviewer ZY6B [Part 2/2]**
>
> ## Questions
> > 1. *I think if possible, the authors can have more discussions on the novelty of this paper.*
>
> We thank the reviewer for their request to elaborate further on the novelty of our work. As mentioned in the related works section of the paper, our work is the first to explicitly formulate both the pre-training and fine-tuning stages within SSL as a BLO problem, which previously have not been explored in SSL research. Current SSL works that integrate BLO, do so in very different manners, utilizing it exclusively in either the pretext or fine-tuning stages, **but not jointly**. Additionally, we refer to our response to Weakness 1 for additional emphasis on the novelty of BiSSL through a discussion on how it differs from the implementation of BLO in meta-learning. This discussion has also been added to the appendix of the paper.
>
> > 2. *I am curious about the results of the proposed method in larger backbone (at least ResNet-50).*
>
> We have addressed the reviewers request, and extended the experiments accordingly. These experiments are also introduced and discussed in our response to a previous review; however, we reiterate them here for the sake of clarity. We conducted the experiments on a selection of downstream datasets (Flowers [3], DTD [4] and VOC07 [5]), where the ResNet-18 backbone has been replaced by a ResNet-50. Otherwise, the setup is identical to as in the paper. The results are as follows:
>
> | Dataset | Top-1/$\ast$mAP (BiSSL) | Top-1/$\ast$mAP (Only FT) |    Avg Diff     |     Top-5 (BiSSL)      | Top-5 (Only FT) |    Avg Diff     |
> | :------ | :------------------------: | :--------------------------: | :-------------: | :--------------------: | :-------------: | :-------------: |
> | Flowers |       $77.7\pm 0.3$        |        $77.1 \pm 0.3$        |     $+0.6$      |     $91.3\pm 0.2$      | $91.0 \pm 0.3$  |     $+0.3$      |
> | DTD     |   $\mathbf{55.4\pm 0.5}$   |        $51.7 \pm 0.5$        | $\mathbf{+3.7}$ | $\mathbf{81.5\pm 0.4}$ | $80.2 \pm 0.5$  | $\mathbf{+1.3}$ |
> | VOC07   |  $*\mathbf{64.0\pm 0.1}$   |       $*62.7 \pm 0.1$        | $\mathbf{+1.3}$ |          $-$           |       $-$       |       $-$       |
>
> The results indicate that BiSSL still outperforms the baselines, despite using a larger backbone model. We hope this addresses the reviewer’s request and that it supports the idea that BiSSL is able to scale well to larger model architectures. We are currently expanding the evaluation to include additional downstream datasets to provide a more comprehensive assessment. We intend to include these results in the appendix of the submission after the review period, due to resource constraints that prevents us from completing these experiments within the rebuttal period.
>
> > 3. *The authors can have more discussions on the limitation of the proposed method.*
>
> Compared to the conventional SSL setup, BiSSL requires the additional access to the pre-training data or a subset of it, as well as knowledge of the pretext task used to pre-train the model. Additionally, calculating the upper-level gradient in BiSSL introduces some non-neglebible computational overhead. We refer to our response to reviewer kv92 for further discussion on this, as well as a quantitative overview of the GPU hours used in the experiments.
>
> We have updated the future work section of the submission to address these limitations as potential areas for further exploration.
>
> ---
>
> ## Conclusion
> **Once again, we thank the reviewer for their comments. We hope our response provides clarity and addresses the concerns raised. We are happy to further elaborate or answer further inquiries as needed.**
>
> ---
>
> ## References
>
> [1] Chelsea Finn, P, et al. "Model-agnostic meta-learning for fast adaptation of deep networks." International Conference on Machine Learning, 2017.
>
> [2] Aravind Rajeswaran, et al. "Meta-learning with im-plicit gradients." Neural Information Processing Systems, 2019.
>
> [3] Maria-Elena Nilsback and Andrew Zisserman. "Automated flower classification over a large number of classes." Indian Conference on Computer Vision, Graphics and Image Processing, Dec 2008.
>
> [4] M. Cimpoi, S. Maji, et al. "Describing textures in the wild." Proceedings of the IEEE Conf. on Computer Vision and Pattern Recognition (CVPR), 2014.
>
> [5] M. Everingham, L. et al. "The PASCAL Visual Object Classes Challenge 2007 (VOC2007)"

---

> ### Author Response · Authors · 2024-11-22
> **Gentle reminder to reviewer ZY6B**
>
> Dear Reviewer ZY6B,
>
> We have responded to your constructive review, incorporating your suggestions as well as including additional experiments to address the concerns and suggestions raised. As the end of the discussion period comes closer, we kindly ask you to read our response. We are happy to provide further clarifications or discuss any further questions you may have.
>
> Best,
>
> Authors

---

> > ### Comment · Reviewer_ZY6B · 2024-11-23
> > **Final decision**
> >
> > After reading other reviews and the rebuttal, I hold my score.

---

> > > ### Author Response · Authors · 2024-11-26
> > >
> > > Thank you so much for taking the time to review our rebuttal and for considering the other reviews!
> > >
> > > Best,
> > > Authors

---

### Official Review · Reviewer_NGXe · 2024-11-03

**Soundness:** 2
**Presentation:** 3
**Contribution:** 2
**Rating:** 3
**Confidence:** 5

**Summary:**

The paper introduces BiSSL, a novel framework leveraging bilevel optimization to enhance self-supervised learning (SSL) by aligning the pre-training and fine-tuning phases. In the conventional SSL process, these stages are typically treated as separate, which can lead to misalignment issues between representations learned during pre-training and those required for downstream tasks. BiSSL addresses this by treating pretext pre-training as a lower-level optimization problem and downstream fine-tuning as an upper-level one, facilitating stronger information flow between stages. Empirical validation using a ResNet-18 architecture and the SimCLR approach, showing improved or competitive accuracy on multiple datasets compared to standard SSL techniques.

**Strengths:**

1. The paper is well-written and easy to follow.

2. The idea of using bilevel optimization for SSL is novel and interesting.

**Weaknesses:**

1. This study relies solely on SimCLR as a comparison method, lacking sufficient comparative experiments with approaches better suited for small-scale datasets, such as BYOL[1] and SwAV[2]. Additionally, it omits necessary explanations of more recent self-supervised methods like DINOv2 [3].

2. There is a lack of comparative experiments on different network architectures, as well as the results of SSL's standard dataset ImageNet. In the absence of sufficient data, the performance of SSL is not obvious compared to SFL.

[1] Grill, Jean-Bastien, et al. "Bootstrap your own latent-a new approach to self-supervised learning." Advances in neural information processing systems 33 (2020): 21271-21284.

[2] Caron, Mathilde, et al. "Unsupervised learning of visual features by contrasting cluster assignments." Advances in neural information processing systems 33 (2020): 9912-9924.

[3] Oquab, Maxime, et al. "Dinov2: Learning robust visual features without supervision." arXiv preprint arXiv:2304.07193 (2023).

**Questions:**

For Fig. 2, did the author analyze why the proposed method converges quickly in the early stages of training? Is there over-fitting?

---

> ### Author Response · Authors · 2024-11-20
> **Author's Response to Reviewer NGXe [Part 1/2]**
>
> We thank the reviewer for their constructive feedback and insightful comments. Below, we address the raised comments.
>
> ---
>
> ## Strengths
> > 1. *The paper is well-written and easy to follow.*
>
> We are happy to hear that the clarity and structure of the paper is appreciated.
>
> > 2. *The idea of using bilevel optimization for SSL is novel and interesting.*
>
> It is gratifying to see that the novelty of our approach is recognized, and we are encouraged by the reviewers interest!
>
> ---
>
> ## Weaknesses
> > 1. *This study relies solely on SimCLR as a comparison method, lacking sufficient comparative experiments with approaches better suited for small-scale datasets, such as BYOL and SwAV. Additionally, it omits necessary explanations of more recent self-supervised methods like DINOv2.*
>
> We agree that expanding our comparisons to include other SSL pre-training methods strengthens the experimental evaluation. To address this, we have conducted additional experiments using BYOL [1] as the pre-training method, and evaluated performance on a selection of the downstream datasets evaluated in the paper (Oxford 102 Flowers [2], DTD [3], and VOC07 [4]). In this experimental setup the predictor network uses an architecture identical to the projector network, and the EMA decay follows a scheduler as in [1] with a base decay of $\tau_{\text{{base}}}=0.9995$ and the base learning rate is additional adjusted to $1.0$, aligning the experimental setup more closely with [1]. Otherwise the setup remains consistent with the one in the paper. The results are presented below:
>
> | Dataset | Top-1/$\ast$mAP (BiSSL) | Top-1/$\ast$mAP (Only FT) |    Avg Diff     |     Top-5 (BiSSL)      | Top-5 (Only FT) |    Avg Diff     |
> | :------ | :------------------------: | :--------------------------: | :-------------: | :--------------------: | :-------------: | :-------------: |
> | Flowers |   $\mathbf{68.4\pm 0.4}$   |        $63.5 \pm 0.8$        | $\mathbf{+4.9}$ | $\mathbf{86.0\pm 0.4}$ | $83.3 \pm 0.7$  | $\mathbf{+3.7}$ |
> | DTD     |       $44.4\pm 0.6$        |        $44.0 \pm 0.7$        |     $+0.4$      | $\mathbf{72.8\pm 0.5}$ | $71.6 \pm 0.5$  | $\mathbf{+1.2}$ |
> | VOC07   |  $*\mathbf{54.2\pm 0.1}$   |       $*51.8 \pm 0.2$        | $\mathbf{+2.4}$ |          $-$           |       $-$       |       $-$       |
>
> We observe that BiSSL in general demonstrate improved downstream performance of the baselines, suggesting that the benefits of BiSSL is not necessarily confined to the pre-training method being SimCLR. While BYOL is expected to be better suited than SimCLR on small-scale datasets, the observed decline in performance could be attributed to differences between our experimental setup and the original BYOL implementation [1] (e.g., size of MLP, uniform weight decay on all layers, hyper-parameter adjustments to dataset, etc.).
>
> Regarding addressing more recent SSL methods, such as DINOv2, we acknowledge this point and have updated our introduction to ensure a more up-to-date overview of the known impactful SSL methods, including DINOv2.
>
> > 2. *There is a lack of comparative experiments on different network architectures, as well as the results of SSL's standard dataset ImageNet. In the absence of sufficient data, the performance of SSL is not obvious compared to SFL.*
>
> We recognize that additional exploration with larger models and pre-training datasets would further enhance the robustness of our findings. To address the reviewer's concerns, we conducted two additional sets of experiments. These experiments are also outlined in our response to a previous review, but we present them here again for the sake of clarity. First we replaced the ResNet-18 backbone used in our initial experiments with a larger ResNet-50 backbone to assess whether BiSSL provides performance benefits on larger architectures. Results are summarized in the table below:
>
> | Dataset | Top-1/$\ast$mAP (BiSSL) | Top-1/$\ast$mAP (Only FT) |    Avg Diff     |     Top-5 (BiSSL)      | Top-5 (Only FT) |    Avg Diff     |
> | :------ | :------------------------: | :--------------------------: | :-------------: | :--------------------: | :-------------: | :-------------: |
> | Flowers |       $77.7\pm 0.3$        |        $77.1 \pm 0.3$        |     $+0.6$      |     $91.3\pm 0.2$      | $91.0 \pm 0.3$  |     $+0.3$      |
> | DTD     |   $\mathbf{55.4\pm 0.5}$   |        $51.7 \pm 0.5$        | $\mathbf{+3.7}$ | $\mathbf{81.5\pm 0.4}$ | $80.2 \pm 0.5$  | $\mathbf{+1.3}$ |
> | VOC07   |  $*\mathbf{64.0\pm 0.1}$   |       $*62.7 \pm 0.1$        | $\mathbf{+1.3}$ |          $-$           |       $-$       |       $-$       |
> *Continued...*

---

> ### Author Response · Authors · 2024-11-20
> **Author's Response to Reviewer NGXe [Part 2/2]**
>
> *Continued answer to W2...*
>
> For the second set of experiments we instead pre-trained on the Imagenet dataset for 100 epochs. Since the BiSSL stage now involves the equivalent of only 8 pretext epochs, we consider the difference in training duration between the baseline and BiSSL setups to be negligible. Hence we utilized the same pre-trained backbone for both the baseline and BiSSL-trained models in these experiments. All other experimental settings remain consistent with those outlined in the paper. The results are as follows:
>
> | Dataset | Top-1/$\ast$mAP (BiSSL) | Top-1/$\ast$mAP (Only FT) |    Avg Diff     |      Top-5 (BiSSL)      | Top-5 (Only FT) |    Avg Diff     |
> | :------ | :------------------------: | :--------------------------: | :-------------: | :---------------------: | :-------------: | :-------------: |
> | Flowers |  $\mathbf{78.8 \pm 0.2}$   |        $77.0 \pm 0.3$        | $\mathbf{+1.8}$ |     $91.6 \pm 0.2$      | $91.9 \pm 0.1$  |     $-0.3$      |
> | DTD     |  $\mathbf{56.7 \pm 0.4}$   |        $53.3 \pm 0.3$        | $\mathbf{+3.4}$ | $\mathbf{83.6 \pm 0.4}$ | $82.3 \pm 0.5$  | $\mathbf{+1.3}$ |
> | VOC07   |  $*\mathbf{62.6 \pm 0.2}$  |       $*60.6 \pm 0.2$        | $\mathbf{+2.0}$ |           $-$           |       $-$       |       $-$       |
>
> In both instances, BiSSL consistently surpasses the baselines, suggesting that its advantages persist even when applied to larger datasets and models. We hope this helps to alleviate the reviewer’s concerns regarding the execution of comparative experiments on larger models and datasets. Should additional architectures or datasets be of interest, we would gladly experiment with such, provided that time allows.
>
> We are currently working on extending all three sets of experiments presented here to include evaluation on a larger selection of downstream datasets. While resource constraints prevent us from obtaining all desired results within the rebuttal period, we will update the paper to include these findings in the appendix as soon as they become available.
>
> ---
>
> ## Questions
> > *For Fig. 2, did the author analyze why the proposed method converges quickly in the early stages of training? Is there over-fitting?*
>
> Thank you for raising this question. To clarify, Fig. 2 does not show the progression of a single model’s performance during training. Instead, it illustrates the final **test** accuracy achieved by separately trained models with varying durations of self-supervised pre-training followed by the application of BiSSL or fine-tuning. The general trend observed in the figure thus indicates that BiSSL consistently retains its classification accuracy advantage, irrespective of the pre-training duration.
>
> We have updated the submission to include a revised description of the experimental setup underlying the figure, along with an updated figure caption.
>
> ---
>
> ## Conlusion
> **We again thank the reviewer for their useful feedback. We hope our responses address the raised concerns, and we remain happy to provide further clarifications or answer additional questions**
>
> ---
>
> ## References
>
> [1] Jean-Bastien Grill, et al. "Bootstrap your own latent-a new approach to self-supervised learning." Advances in neural information processing systems, 33:21271–21284, 2020.
>
> [2] Maria-Elena Nilsback and Andrew Zisserman. "Automated flower classification over a large number of classes." Indian Conference on Computer Vision, Graphics and Image Processing, Dec 2008.
>
> [3] M. Cimpoi, S. Maji, et al. "Describing textures in the wild." Proceedings of the IEEE Conf. on Computer Vision and Pattern Recognition (CVPR), 2014.
>
> [4] M. Everingham, L. et al. "The PASCAL Visual Object Classes Challenge 2007 (VOC2007)"

---

> ### Author Response · Authors · 2024-11-22
> **A kind nudge to reviewer NGXe**
>
> Dear Reviewer NGXe,
>
> Thank you again for your constructive feedback. We have addressed your concerns and included additional experiments as part of our response. With the end of the discussion period slowly approaching, we would appreciate if you could read our response so we can address any further questions that you might have.
>
> Best,
>
> Authors

---

> > ### Comment · Reviewer_NGXe · 2024-11-26
> >
> > I appreciate the author's detailed response and additional experiments, but considering the opinions of other reviewers and the fact that the method is not general enough, and most importantly, most of the key experiments were supplemented after the review, I think it is necessary to revise it again to meet the requirement, so I decided to maintain my score.

---

> > > ### Author Response · Authors · 2024-11-26
> > >
> > > Thank you for your feedback and for acknowledging the substance of our response. We greatly value your perspective and appreciate the time you have dedicated to reviewing our work.
> > >
> > > Regarding your statement that BiSSL is not general enough, we would like to emphasize that BiSSL is explicitly designed to be broadly applicable. The theoretical formulation underlying BiSSL is intentionally crafted to generally apply to almost any training setup that follows the conventional self-supervised pre-training and fine-tuning pipeline. The sole critical assumption is that the pretext and downstream loss functions are differentiable, which is a standard property of most deep learning tasks. Consequently, BiSSL is compatible with almost all known common pretext and downstream tasks. Additionally, in setups where pre-training and fine-tuning are already implemented, BiSSL can with little effort be incorporated as a plug-and-play enhancement, as further elaborated in our response to another review. If there are specific limitations you foresee in its generalizability, we would be eager to understand them further.
> > >
> > > Regarding the experiments, we argue that the key experimental setup and results are those detailed in the original submission, as they demonstrate the core conclusion that BiSSL can improve downstream performance over conventional fine-tuning by better aligning the self-supervised pre-trained backbone with the downstream task. The supplementary experiments provided during the rebuttal were slight variations of the original setup, conducted to address reviewer concerns, and primarily restated the claims of BiSSL's benefits.
> > >
> > > Again we appreciate your feedback and remain open to further discussion.

---

### Official Review · Reviewer_kv92 · 2024-11-03

**Soundness:** 3
**Presentation:** 3
**Contribution:** 2
**Rating:** 5
**Confidence:** 3

**Summary:**

The paper introduces BiSSL, a training framework that integrates bilevel optimization into the self-supervised learning pipeline to enhance the alignment between pretraining and downstream fine-tuning stages. BiSSL formulates the pretext and downstream task objectives as lower- and upper-level objectives within a bilevel optimization problem, facilitating improved information sharing between the stages and leading to better parameter initialization for downstream tasks. Results in image classification datasets show the effectiveness of the proposed method.

**Strengths:**

1. Introduces a new training framework that leverages bilevel optimization to bridge the gap between pretext pretraining and downstream tasks.

2. Demonstrates improved or comparable classification accuracies across multiple image classification datasets compared to traditional SSL methods. The author also  provide insights into how BiSSL affects the learned representations, showing enhanced downstream feature alignment before fine-tuning.

**Weaknesses:**

* **Limited Experimental Scope** : The evaluation of BiSSL is confined to classification tasks, and the models and datasets used are relatively small-scale. This restricts the demonstration of the framework's scalability and its ability to handle larger and more complex datasets, which is a critical aspect for real-world applications.
* **Compromise on Generalizability** : The pretraining setup of BiSSL aims to align with specific downstream tasks, which might sacrifice the generalizability of the pretrained models. This alignment could potentially limit the framework's effectiveness for a diverse range of tasks that were not considered during the pretraining phase.
* **Increased Training Complexity** : The introduction of bilevel optimization makes the training process more complex compared to the traditional pretraining and downstream fine-tuning paradigm. This added complexity could pose challenges in terms of computational resources and may require additional expertise to implement and optimize, potentially hindering its adoption in simpler or more streamlined workflows.

**Questions:**

see Weaknesses

---

> ### Author Response · Authors · 2024-11-19
> **Author's Response to Reviewer kv92 [Part 1/3]**
>
> We thank the reviewer for their comments. We are very pleased that the novelty and demonstrations of the BiSSL framework has been recognized!
>
> ## Weaknesses  + Questions
> ### Weakness/Question 1
> > **Limited Experimental Scope**: *The evaluation of BiSSL is confined to classification tasks, and the models and datasets used are relatively small-scale. This restricts the demonstration of the framework's scalability and its ability to handle larger and more complex datasets, which is a critical aspect for real-world applications.*
>
> We acknowledge the reviewer’s concern regarding the scale of the datasets used in our evaluation. The downstream datasets in our experiments, such as VOC07 [1] and DTD [2], are commonly used to benchmark self-supervised learning methods, including works like SimCLR [3] and BYOL [4]. Hence, we argue that these benchmarks provide a meaningful basis for downstream evaluation. However, while the STL10 dataset used for pre-training in our experiments provides reasonable scale and complexity, we acknowledge the importance of experimenting whether the benefits from BiSSL persists with larger-scale pre-training datasets like ImageNet. To address this limitation, we have extended the experiments using ImageNet for pre-training. In these experiments, we conducted pre-training on Imagenet for 100 epochs, using the same pre-trained backbone for both the BiSSL and baseline models. The following table outlines downstream classification accuracies on a selection of the downstream datasets utilised in the experiments of the paper:
>
> | Dataset     | Top-1/$\ast$mAP (BiSSL) | Top-1/$\ast$mAP (Only FT) |    Avg Diff     |      Top-5 (BiSSL)      | Top-5 (Only FT) |    Avg Diff     |
> | :---------- | :------------------------: | :--------------------------: | :-------------: | :---------------------: | :-------------: | :-------------: |
> | Flowers [5] |  $\mathbf{78.8 \pm 0.2}$   |        $77.0 \pm 0.3$        | $\mathbf{+1.8}$ |     $91.6 \pm 0.2$      | $91.9 \pm 0.1$  |     $-0.3$      |
> | DTD         |  $\mathbf{56.7 \pm 0.4}$   |        $53.3 \pm 0.3$        | $\mathbf{+3.4}$ | $\mathbf{83.6 \pm 0.4}$ | $82.3 \pm 0.5$  | $\mathbf{+1.3}$ |
> | VOC07       |  $*\mathbf{62.6 \pm 0.2}$  |       $*60.6 \pm 0.2$        | $\mathbf{+2.0}$ |           $-$           |       $-$       |       $-$       |
>
> Additionally, to address the similar concerns regarding experiments using larger-scale models, we repeated the same experiments as in the paper, but with a larger ResNet-50 instead of a ResNet-18 as the backbone model:
>
> | Dataset | Top-1/$\ast$mAP (BiSSL) | Top-1/$\ast$mAP (Only FT) |    Avg Diff     |     Top-5 (BiSSL)      | Top-5 (Only FT) |    Avg Diff     |
> | :------ | :------------------------: | :--------------------------: | :-------------: | :--------------------: | :-------------: | :-------------: |
> | Flowers |       $77.7\pm 0.3$        |        $77.1 \pm 0.3$        |     $+0.6$      |     $91.3\pm 0.2$      | $91.0 \pm 0.3$  |     $+0.3$      |
> | DTD     |   $\mathbf{55.4\pm 0.5}$   |   $51.7 \pm 0.5$    | $\mathbf{+3.7}$ | $\mathbf{81.5\pm 0.4}$ | $80.2 \pm 0.5$  | $\mathbf{+1.3}$ |
> | VOC07   |  $*\mathbf{64.0\pm 0.1}$   |       $*62.7 \pm 0.1$        | $\mathbf{+1.3}$ |          $-$           |       $-$       |       $-$       |
>
> In both cases, BiSSL generally outperforms the baselines, implying that its benefits remain present on larger datasets and models. We hope this additional evidence addresses the concern regarding BiSSL’s applicability on larger datasets and models. Furthermore, we are in the process of expanding both sets of experiments to encompass a wider range of downstream datasets. Although computational resource limitations prevent us from obtaining all desired results within the review period, we will incorporate these findings into the appendix of the paper as soon as they become available.
>
> *Continued...*

---

> ### Author Response · Authors · 2024-11-19
> **Author's Response to Reviewer kv92 [Part 2/3]**
>
> ### Weakness/Question 2
> > **Compromise on Generalizability**: *The pre-training setup of BiSSL aims to align with specific downstream tasks, which might sacrifice the generalizability of the pre-trained models. This alignment could potentially limit the framework's effectiveness for a diverse range of tasks that were not considered during the pre-training phase.*
>
> While BiSSL may limit the backbone’s applicability to other tasks, its primary objective is not to create a general-purpose model. Rather, the goal is to enhance the performance of a pre-trained backbone on a specific downstream task, akin to the purpose of fine-tuning in general. This is also reflected in our training pipeline, which initiates BiSSL with a general-purpose backbone obtained through self-supervised pre-training. To disclose, we did in our experiments opt for shorter pre-training durations before applying BiSSL to facilitate more direct comparisons with the baselines. However, this does not suggest that BiSSL inherently requires less pre-training to be effective. Practitioners can apply BiSSL to any sufficiently pre-trained backbone, as evidenced in Fig. 2 of the paper, where BiSSL consistently outperforms the baseline once pre-training exceeds a certain duration.
>
> We appreciate the opportunity to clarify this point and hope this explanation addresses the reviewer’s comment. We are happy to elaborate further if desired.
>
> ### Weakness/Question 3
> > **Increased Training Complexity**: *The introduction of bilevel optimization makes the training process more complex compared to the traditional pre-training and downstream fine-tuning paradigm. This added complexity could pose challenges in terms of computational resources and may require additional expertise to implement and optimize, potentially hindering its adoption in simpler or more streamlined workflows.*
>
> We acknowledge that bilevel optimization introduces additional computational steps compared to the conventional self-supervised training pipeline. To provide transparency regarding the computational requirements, we document the GPU hours required for pretext training and BiSSL for the various documented experimental setups:
>
> | (Pre-Training Dataset, Backbone, GPUs used) | Conventional Pipeline | Pipeline with BiSSL |
> | :------------------------------------------ | :-------------------: | :-----------------: |
> | Pretext Training (STL10, R18, 4×A40)        |     64 GPU Hours      |    52 GPU Hours     |
> | BiSSL Training (STL10, R18, 4×A40)          |           -           |    20 GPU Hours     |
> |                                             |                       |                     |
> | Pretext Training (STL10, R50, 4×A40)        |     80 GPU Hours      |    67 GPU Hours     |
> | BiSSL Training (STL10, R50, 4×A40)          |           -           |   63.5 GPU Hours    |
> |                                             |                       |                     |
> | Pretext Training (ImageNet, R18, 8×A100)    |     424 GPU Hours     |    424 GPU Hours    |
> | BiSSL Training (ImageNet, R18, 4×A40)       |           -           |   23.5 GPU Hours    |
>
> While the computational cost of BiSSL increases with model size, it remains relatively constant with respect to the size of the pre-training dataset, as the number of gradient steps for BiSSL training is fixed by design. We aim to include this table in the appendix once the aforementioned experimental results are available. Additionally, the conjugate gradient algorithm used in our implementation is not the most recent or efficient approach for calculating upper-level gradients in bilevel optimization. Future work could explore more recent techniques [6,7] to reduce BiSSL’s computational overhead without compromising performance. We updated the submission to reflect this as an opportunity for future work.
>
> Regarding implementation complexity, BiSSL is designed to be plug-and-play, requiring minimal modifications to existing pre-training and fine-tuning workflows. Any practitioner familiar with SSL and fine-tuning should find it straightforward to integrate BiSSL into their pipeline. To lower the barrier to adoption, we have made our implementation open-source and [publicly available on GitHub](https://github.com/ICLR25-10484/ICLR25_10484_BiSSL), accompanied by documentation and examples.
>
> We hope this demonstrates that while BiSSL introduces some additional complexity, it remains feasible for broader adoption.
>
> ---
> ## Conclusion
> **We thank the reviewer again for their constructive feedback. We hope our responses address these concerns, and we remain happy to provide further clarifications or answer additional questions.**

---

> ### Author Response · Authors · 2024-11-19
> **Author's Response to Reviewer kv92 [Part 3/3]**
>
> ## References
>
> [1] M. Everingham, L. et al. "The PASCAL Visual Object Classes Challenge 2007 (VOC2007)"
>
> [2] M. Cimpoi, S. Maji, et al. "Describing textures in the wild." Proceedings of the IEEE Conf. on Computer Vision and Pattern Recognition (CVPR), 2014.
>
> [3] Ting Chen, et al. "A simple framework for contrastive learning of visual representations." International conference on machine learning, pp. 1597–1607. PMLR, 2020.
>
> [4] Jean-Bastien Grill, et al. "Bootstrap your own latent-a new approach to self-supervised learning." Advances in neural information processing systems, 33:21271–21284, 2020.
>
> [5] Maria-Elena Nilsback and Andrew Zisserman. "Automated flower classification over a large number of classes." Indian Conference on Computer Vision, Graphics and Image Processing, Dec 2008.
>
> [6] Junjie Yang, et al. "Provably faster algorithms for bilevel optimization."  Advances in Neural Information Processing Systems, 2021.
>
> [7] Feihu Huang. "Optimal Hessian/Jacobian-free nonconvex-PL bilevel optimization." Proceedings of the 41st International Conference on Machine Learning, volume 235 of Proceedings of Machine Learning Research, pp. 19598–19621. PMLR, 21–27, Jul 2024.

---

> ### Author Response · Authors · 2024-11-22
> **A gentle follow-up to reviewer kv92**
>
> Dear Reviewer kv92,
>
> We have responded to your review, covering your concerns and adding new experiments. As the end of the discussion period is approaching us, we would greatly appreciate it if you could review our response and let us know if there are any further questions or clarifications needed.
>
> Best,
>
> Authors

---

> > ### Comment · Reviewer_kv92 · 2024-11-25
> >
> > After considering the other reviews and the rebuttal, I lean towards maintaining the current rating. As mentioned by authors  that “BiSSL may limit the backbone’s applicability to other tasks, its primary objective is not to create a general-purpose model”, which reduces the significance of the paper.

---

> ### Author Response · Authors · 2024-11-25
> **Comment to Reviewer's Response: Clarifying the Focus of BiSSL + New Experiments**
>
> Thank you for considering our rebuttal. We would like to clarify the intent of the statement you highlighted. That particular statement was intended to explain that BiSSL is a fine-tuning method rather than a pre-training approach. When deploying a pre-trained backbone for a specific downstream application, fine-tuning is applied. After fine-tuning for a specific task, the backbone’s applicability to other tasks may naturally be limited. Fine-tuning is a critical step in self-supervised learning pipelines, and BiSSL enhances this process, as evidenced by its consistent performance gains over baselines in our experiments. To the best of our knowledge, BiSSL is the first to explicitly incorporate knowledge of the pre-training objective in conjunction with downstream fine-tuning.
>
> In the meantime, we have conducted additional experiments to further address the reviewer’s comments regarding the use of larger models and datasets. Specifically, we used a ResNet-50 backbone pre-trained on ImageNet, which aligns closely with the experimental scope of SSL works such as SimCLR and BYOL. The results are as follows:
>
> | Dataset | Top-1 / $\ast$ mAP (BiSSL) | Top-1 / $\ast$ mAP (Only FT) |    Avg Diff     |      Top-5 (BiSSL)      | Top-5 (Only FT) |    Avg Diff     |
> | :------ | :------------------------: | :--------------------------: | :-------------: | :---------------------: | :-------------: | :-------------: |
> | Flowers |  $\mathbf{81.4 \pm 0.4}$   |        $80.3 \pm 0.5$        | $\mathbf{+1.1}$ |     $93.1 \pm 0.3$      | $93.3 \pm 0.3$  |     $-0.2$      |
> | DTD     |  $\mathbf{60.7 \pm 0.4}$   |        $57.6 \pm 0.3$        | $\mathbf{+3.1}$ | $\mathbf{86.3 \pm 0.5}$ | $84.0 \pm 0.4$  | $\mathbf{+2.3}$ |
> | VOC07   |  $*\mathbf{68.4 \pm 0.1}$  |       $*67.7 \pm 0.1$        | $\mathbf{+0.7}$ |           $-$           |       $-$       |       $-$       |
>
> Once again, BiSSL demonstrates superior performance compared to conventional fine-tuning.

---

### Author Response · Authors · 2024-11-21
**Overview and Overall Response**

We thank the reviewers for taking the time to review our paper and provide valuable feedback. We are delighted with their acknowledgment of the paper's novelty as well as their positive comments on its writing and organization. Below, we provide a summary of our responses, including the new experiments conducted.

---

# Experimental Scope
A common theme among the reviews pertained to the experimental scope, including the request for evaluating on larger backbones, pre-training datasets, and alternative self-supervised learning frameworks. To address these concerns, we conducted four additional experiments. Each experiment involved one specific change to the original setup while keeping all other elements consistent with the experiments of the paper:
- **Experiment 1:** Exchanging the previous ResNet-18 backbone with a larger ResNet-50.
- **Experiment 2:** Exchanging the pre-training dataset from STL10 to the larger-scale ImageNet dataset.
- **Experiment 3:** Exchanging the previous ResNet-18 backbone with a larger ResNet-50 + Exchanging the pre-training dataset from STL10 to the larger-scale ImageNet dataset.
- **Experiment 4:** Using Bootstrap Your Own Latent (BYOL) for self-supervised pre-training instead of SimCLR.

We evaluated model performance in the same manner as in the paper, using a selection of the downstream datasets used in the paper. We refer to our responses to the individual reviews for the results. Across all experiments, BiSSL still demonstrated significant improvements on a majority of the datasets, while showing comparable performance on the remaining, further highlighting its flexibility across architectures, datasets, and SSL methods.

We are working to expand the evaluation of these experiments across a larger selection of downstream datasets. However, due to resource constraints, we were unable to complete these within the rebuttal period, but we will include them in the final submission.


# Discussion of Limitations
Reviewers kv92 and ZY6B raised questions about BiSSL's limitations. We highlighted its increased training complexity and the need for access to pre-training data and pretext tasks, including these points in the submission. For additional transparency, we documented computation times for both BiSSL and the baselines, referring to our response to Reviewer kv92 for details. The same reviewer also raised concerns about implementing BiSSL in practical setups, which we addressed by clarifying its plug-and-play design for practitioners familiar with SSL workflows. To further ease adoption, we have [open sourced the code](https://github.com/ICLR25-10484/ICLR25_10484_BiSSL).

---

# Conclusion
**We hope our responses and updates have adequately address the reviewers’ concerns. We are grateful for their constructive feedback and remain happy to provide further clarifications or answer additional questions.**

---

### Meta-Review · Area_Chair_ddwM · 2024-12-16

**Metareview:**

This paper proposes a bilevel optimization of SSL and fine-tuning on ResNet18 and STL10 dataset. It received mixed reviews where [kv92, NGXe] are negative while [ZY6B] is positive. The reviewers feel bilevel optimization is generally interesting, and the presentation is mostly clear. The downside is that lack of comparison and generalizations. Overall, reviewers are converged to the limited applicability to different tasks, which decreases the significance. The AC has checked all files and stand on the reviewers' side. The authors are suggested to further improve the current work.

**Additional Comments On Reviewer Discussion:**

[kv92] raised limited experimental scope, compromise on generalizability, and increased training complexity. These points are addressed by providing more experiments and detailed explanations. [NGXe] raised lacking sufficient comparative experiments issues regarding the method and network backbones, which are addressed by providing additional experiments. [ZY6B] raised limited technical novelty, small backbone, which are detailedly explained by the authors.

---

### Decision · Program_Chairs · 2025-01-22

Reject